# Towards Empowerment Gain through Causal Structure Learning in Model-Based Reinforcement Learning

**Hongye Cao**[1*]    **Fan Feng**[2,3*]    **Meng Fang**[4]    **Shaokang Dong**[1]    **Tianpei Yang**[1,5]
**Jing Huo**[1]    **Yang Gao**[1,5]
[1]National Key Laboratory for Novel Software Technology, Nanjing University
[2]University of California, San Diego    [3]MBZUAI    [4]University of Liverpool
[5]School of Intelligence Science and Technology, Nanjing University

## Abstract

In Model-Based Reinforcement Learning (MBRL), incorporating causal structures into dynamics models provides agents with the structured understanding of environments, enabling more efficient and effective decisions. Empowerment, as an intrinsic motivation, enhances the ability of agents to actively control environments by maximizing mutual information between future states and actions. We posit that empowerment coupled with the causal understanding of the environment can improve the agent's controllability over environments, while enhanced empowerment gain can further facilitate causal reasoning. To this end, we propose the framework that pioneers the integration of empowerment with causal reasoning, Empowerment through Causal Learning (**ECL**), where an agent with the awareness of the causal dynamics model achieves empowerment-driven exploration and optimizes its causal structure for task learning. Specifically, we first train a causal dynamics model of the environment based on collected data. Next, we maximize empowerment under the causal structure for exploration, simultaneously using data gathered through exploration to update the causal dynamics model, which could be more controllable than dynamics models without the causal structure. We also design an intrinsic curiosity reward to mitigate overfitting during downstream task learning. Importantly, **ECL** is method-agnostic and can integrate diverse causal discovery methods. We evaluate **ECL** combined with 3 causal discovery methods across 6 environments including both state-based and pixel-based tasks, demonstrating its performance gain compared to other causal MBRL methods, in terms of causal structure discovery, sample efficiency, and asymptotic performance in policy learning. The project page is https://sites.google.com/view/ecl-1429/.

## 1 Introduction

Model-Based Reinforcement Learning (MBRL) uses predictive dynamics models to enhance decision-making and planning (Moerland et al., 2023). Recent advances in integrating causal structures into MBRL have provided a more accurate description of systems, achieve better adaptation (Huang et al., 2021; 2022; Feng & Magliacane, 2023), generalization (Pitis et al., 2022; Zhang et al., 2020; Wang et al., 2022c; Richens & Everitt, 2024; Lu et al., 2021), and avoiding spurious correlations (Ding et al., 2022; 2024; Liu et al., 2024; Mutti et al., 2023a).

However, these methods often *passively* rely on pre-existing or learned causal structures for policy learning or generalization. In this work, we aim to enable the agent to *actively* leverage causal structures, guiding more efficient exploration of the environment. The agent can then refine its causal structure through newly acquired data, resulting in improvements in both the causal model and policy. This could further enhance the agent's controllability over the environment and its learning efficiency.

We hypothesize that agents equipped with learned causal structures will have better controllability than those using traditional dynamics models without causal modeling. This is because causal

---

*Equal contribution. Corresponding to Jing Huo (huojing@nju.edu.cn).

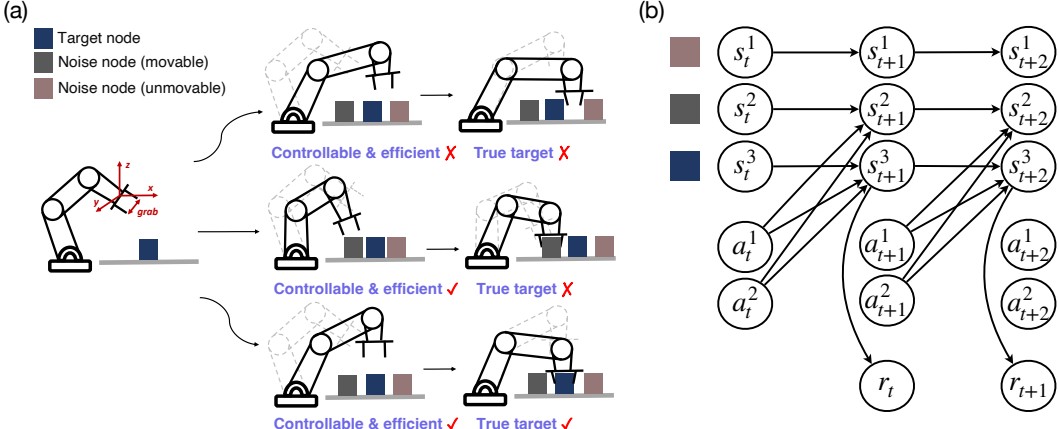

Figure 1: (a). An example of a robot manipulation task with three trajectories and three nodes: one target node (movable) and two noisy nodes (one movable, one unmovable). (b). Underlying causal structures with a factored MDP. Different nodes represent different dimensional states and actions.

structures inform agents to explore the environment more efficiently by nulling out the irrelevant system variables. This assumption serves as intrinsic motivation to guide the policy in exploring higher-quality data, which in turn improves both causal and policy learning. Specifically, we employ empowerment gain, an information-theoretic framework where agents maximize mutual information between their actions and future states to improve control (Leibfried et al., 2019; Klyubin et al., 2005; 2008; Bharadhwaj et al., 2022; Eysenbach et al., 2018; Mohamed & Jimenez Rezende, 2015), as the intrinsic motivation to measure the agent's controllability. Concurrently, through empowerment, agents develop a more nuanced comprehension of their actions' consequences, implicitly discovering the causal relationships within their environment. Hence, by iteratively *improving empowerment gain with causal structure for exploration*, *refining causal structure with data gathered through the exploration*, the agent should be able to develop a robust causal model for effective policy learning.

We give a motivating example (Fig.1(a)) in a manipulation task, where the robot aims to move a target node while avoiding noisy nodes. Three possible trajectories (rows 1-3) are shown with different levels of control, efficiency, and success. Row 1 (irrelevant states) represents the least effective trajectory that can not control nodes and find the target, while rows 2 and 3 (controllable states) demonstrate learned control and efficiency, with high empowerment focusing on movable objects. In the corresponding causal graphs, represented as a Dynamics Bayesian Network in Fig. 1(b), $s^1$, $s^2$, $s^3$ denote the states of three objects. For simplicity and clarity, we assume each object is represented by a single variable. The graph illustrates the causal relationships between these states, actions, and rewards. Assuming the agent follows the causal structure (Fig.1(b)), it will likely execute actions similar to rows 2 and 3 since there are causal relationships between actions and states of movable objects, effectively improving controllability. Through exploration with better control, agents can facilitate improved causal discovery of the task, leading to high-reward outcomes and resulting in more efficient task completion like row 3.

To this end, we propose an Empowerment through Causal Learning (`ECL`) framework that *actively* leverages causal structure to maximize empowerment gain, improving controllability and learning efficiency. `ECL` consists of three main steps: model learning, model optimization, and policy learning. In model learning (step 1), we learn the causal dynamics model with a causal mask and a reward model. We then integrate an empowerment-driven exploration policy with the learned causal structure to better control the environment (step 2). We alternately update the causal structure with the collected data through exploration and policy of empowerment maximization. Finally, the optimized causal dynamics and reward models are used to learn policies for downstream tasks with a curiosity reward to maintain robustness and prevent overfitting (step 3). Importantly, `ECL` is method-agnostic, being able to integrate diverse causal discovery (i.e., score-based and constraint-based) methods. The main contributions of this work can be summarized as follows:

- To improve controllability and learning efficiency, we propose `ECL`, a novel method-agnostic framework that actively leverages causal structures to boost empowerment gain, facilitating efficient exploration and causal discovery.

- **ECL** leverages causal dynamics model to conduct empowerment-based exploration. It also utilizes controllable data gathered through exploration to optimize causal structure and reward models, thereby delving deeper into the causal relationships among states, actions, and rewards.
- We evaluate **ECL** combined with 3 causal discovery methods across 6 environments, encompassing both In-Distribution (ID) and Out-Of-Distribution (OOD) settings, as well as pixel-based tasks. Our results demonstrate that **ECL** outperforms other causal MBRL methods, exhibiting superior performance in terms of causal discovery accuracy, sample efficiency, and asymptotic performance.

## 2 PRELIMINARIES

### 2.1 MDP WITH CAUSAL STRUCTURES

**Markov Decision Process**  In MBRL, the interaction between the agent and the environment is formalized as a Markov Decision Process (MDP). The standard MDP is defined by the tuple $\mathcal{M} = \langle \mathcal{S}, \mathcal{A}, T, \mu_0, r, \gamma \rangle$, where $\mathcal{S}$ denotes the state space, $\mathcal{A}$ represents the action space, $T(s'|s, a)$ is the transition dynamics model, $r(s, a)$ is the reward function, and $\mu_0$ is the distribution of the initial state $s_0$. The discount factor $\gamma \in [0, 1)$ is also included. The objective of RL is to learn a policy $\pi : \mathcal{S} \times \mathcal{A} \to [0, 1]$ that maximizes the expected discounted cumulative reward $\eta_{\mathcal{M}}(\pi) := \mathbb{E}_{s_0 \sim \mu_0, s_t \sim T, a_t \sim \pi} \left[ \sum_{t=0}^{\infty} \gamma^t r(s_t, a_t) \right]$.

**Structural Causal Model**  A Structural Causal Model (SCM) (Pearl, 2009) is defined by a distribution over random variables $\mathcal{V} = \{s_t^1, \cdots, s_t^d, a_t^1, \cdots, a_t^n, s_{t+1}^1, \cdots, s_{t+1}^d\}$ and a Directed Acyclic Graph (DAG) $\mathcal{G} = (\mathcal{V}, \mathcal{E})$ with a conditional distribution $P(v_i|\text{PA}(v_i))$ for node $v_i \in \mathcal{V}$. Then the distribution can be specified as:

$$p(v^1, \ldots, v^{|\mathcal{V}|}) = \prod_{i=1}^{|\mathcal{V}|} p(v^i|\text{PA}(v_i)), \tag{1}$$

where $\text{PA}(v_i)$ is the set of parents of the node $v_i$ in the graph $\mathcal{G}$.

**Causal Structures in MDP**  We model a factored MDP (Guestrin et al., 2003; 2001) with the underlying SCM between states, actions, and rewards (Fig.1b). In this factored MDP, nodes represent system variables (different dimensions of the state, action, and reward), while edges denote their relationships within the MDP. We employ causal discovery methods to learn the structures of $\mathcal{G}$. We identify the graph structures in $\mathcal{G}$, which can be represented as the causal mask $M$. Hence, the dynamics transitions and reward functions in MDP with causal structures are defined as follows:

$$\begin{cases} s_{t+1}^i = f\left(M^{s \to s} \odot s_t, M^{a \to s} \odot a_t, \epsilon_{s,i,t}\right) \\ \quad\quad r_t = R(\phi_c(s_t \mid M), a_t) \end{cases} \tag{2}$$

where $s_{t+1}^i$ represents the next state in dimension $i$, $M^{s \to s} \in \{0, 1\}^{|s| \times |s|}$ and $M^{a \to s} \in \{0, 1\}^{|a| \times |s|}$ are the causal masks indicating the influence of current states and actions on the next state, respectively, $\odot$ denotes the element-wise product, and $\epsilon_{s,i,t}$ represents i.i.d. Gaussian noise. Each entry in the causal mask $M$ (represented as the adjacency matrix of the causal graph $\mathcal{G}$) indicates the presence (1) or absence (0) of a causal relationship between elements. The reward $r_t$ is a function of the state abstraction $\phi_c(\cdot \mid M)$ under the learned causal mask $M$, which filters out the state dimensions without direct edges to the target state dimension, and the action $a_t$. We list the assumptions and propositions in Appendix C.

### 2.2 EMPOWERMENT

Empowerment is to quantify the influence an agent has over its environment and the extent to which this influence can be perceived by the agent (Klyubin et al., 2005; Salge et al., 2014; Jung et al., 2011). Within our framework, the empowerment is the mutual information between the agent action $a_t$ and its subsequent state $s_{t+1}$ under the causal mask $M$ as follows:

$$\mathcal{E} := \max_{\pi(\cdot|s_t)} \mathcal{I}(s_{t+1}; a_t \mid M), \tag{3}$$

where $\mathcal{E}$ is used to represent the channel capacity from the action to state observation. $\pi(\cdot|s_t)$ is the conditional distribution of actions given states.

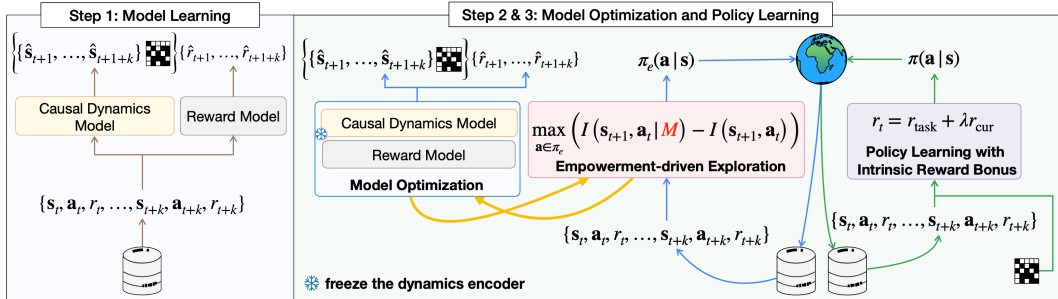

Figure 2: The framework overview of **ECL**. Gold lines: model learning. Blue lines: model optimization alternating with empowerment-driven exploration (yellow lines). Green lines: policy learning.

## 3 EMPOWERMENT THROUGH CAUSAL LEARNING

An illustration of the **ECL** framework is shown in Fig. 2, comprising three main steps: model learning, model optimization, and policy learning. In model learning **(step 1)**, we learn causal dynamics model with the causal mask and reward model. This causal dynamics model is trained using collected data to identify causal structures (i.e., causal masks $M$) , by maximizing the likelihood of observed trajectories. The reward model is trained based on state abstraction that masks irrelevant state dimensions with the causal structure. With the learned causal structure, we integrate empowerment-driven exploration for model optimization **(step 2)**. This process involves learning the empowerment policy $\pi_e$ that enhances the agent's controllability by actively leveraging the causal mask. We alternately update the policy $\pi_e$ for empowerment maximization and generate data with $\pi_e$ to optimize the causal mask $M$ and reward model $P_{\varphi_r}$. Finally, in **step 3**, the learned causal dynamics and reward models are used to learn policies for the downstream tasks. In addition to the task reward, to maintain robustness and prevent overfitting, an intrinsic curiosity reward is incorporated to balance the causality.

### 3.1 STEP 1: MODEL LEARNING WITH CAUSAL DISCOVERY

We first learn causal dynamics model with the causal mask and reward model for the empowerment and downstream task learning. Specifically, a dynamics encoder is trained by maximizing the likelihood of observed trajectories $\mathcal{D}$. Then, the causal mask is learned based on the dynamics model and a reward model is trained with the state abstraction under the causal mask and action.

**Causal Dynamics Model**   The causal dynamics model is composed with a dynamics model $P_{\phi_c}$ and a causal mask $M$. The dynamics model maximizes the likelihood of observed trajectories $\mathcal{D}$ as follows:

$$\mathcal{L}_{\text{dyn}} = \mathbb{E}_{(s_t,a_t,s_{t+1})\sim\mathcal{D}}\left[\sum_{i=1}^{d_S}\log P_{\phi_c}(s_{t+1}^i|s_t,a_t;\phi_c)\right], \tag{4}$$

where $d_S$ is the dimension of the state space, and $\phi_c$ denotes the parameters of the dynamics model. We train the dynamics model as a dense dynamics model that incorporates all state dimensions to capture the state transitions within the environment, facilitating subsequent causal discovery and empowerment. Additionally, we assess the performance of the dense model, specifically the baseline MLP, within the experimental evaluations detailed in Section 5. Next, we use this learned dynamics model for causal discovery.

**Causal Discovery**   For causal discovery, with the learned dynamics model $P_{\phi_c}$, we further embed the causal masks $M^{s\to s}$ and $M^{a\to s}$ into the learning objective. To learn the causal mask, we employ both conditional independence testing (*constraint-based*) (Wang et al., 2022c) and mask learning by sparse regularization (*score-based*) (Huang et al., 2022). We further maximize the likelihood of states by updating the dynamics model and learned masks. Thus, the learning objective for the causal dynamics model is as follows:

$$\mathcal{L}_{\text{c-dyn}} = \mathbb{E}_{(s_t,a_t,s_{t+1})\sim\mathcal{D}}\left[\sum_{i=1}^{d_S}\log P_{\phi_c}(s_{t+1}^i|M^{s\to s^j}\odot s_t, M^{a\to s^j}\odot a_t;\phi_c) + \mathcal{L}_{\text{causal}}\right], \tag{5}$$

where $\mathcal{L}_{\text{causal}}$ represents the objective term associated with learning the causal structure. $\mathcal{L}_{\text{causal}}^{\text{Con}} = \sum_{j=1}^{d_S} \left[ \log \hat{p}(s_{t+1}^j | \{a_t, s_t \setminus s_t^i\}) \right]$ and $\mathcal{L}_{\text{causal}}^{\text{Sco}} = -\lambda_M ||M||_1$ represent constraint-based and score-based objectives respectively. $\lambda_M$ is regularization coefficient.

**Reward Model**   After obtaining the causal dynamics model, we process states using the causal mask $M$ to derive state abstractions $\phi_c(\cdot \mid M)$ for the reward model learning, effectively filtering out irrelevant state dimensions. Simultaneously, the reward model $P_{\varphi_r}$ maximizes the likelihood of observed rewards sampled from trajectories $D$:

$$\mathcal{L}_{\text{rew}} = \mathbb{E}_{(s_t, a_t, r_t) \sim \mathcal{D}} \left[ \log P_{\varphi_r} (r_t | \phi_c(s_t \mid M), a_t) \right]. \tag{6}$$

In this way, `ECL` leverages causal understanding to enhance both state representation and reward prediction accuracy. Finally, the overall objective of the model learning with the causal structure is to maximize $\mathcal{L} = \mathcal{L}_{\text{dyn}} + \mathcal{L}_{\text{c-dyn}} + \mathcal{L}_{\text{rew}}$.

## 3.2   STEP 2: MODEL OPTIMIZATION WITH EMPOWERMENT-DRIVEN EXPLORATION

In Step 2, we optimize the learning of the causal structure and empowerment. As depicted in Fig. 2, this procedure alternates between optimizing the empowerment-driven exploration policy $\pi_e$ and update the causal mask $M$ using data gathered through exploration. Furthermore, to ensure the stability, we update the reward model to adapt to changes in state abstraction induced by updates to the causal mask $M$. Note that the dynamics model $P_{\phi_c}$ learned in Step 1 remains fixed, allowing for a focused optimization of both the causal structure and the empowerment in an alternating manner. The causal structure is optimized by the causal mask M through maximizing $\mathcal{L}_{causal}$ (Eq. 5), while keeping the parameters of $\phi_c$ fixed during this learning step.

**Empowerment-driven Exploration**   To enhance the agent's control and efficiency given the causal structure, instead of maximizing $\mathcal{I}(s_{t+1}, a_t | s_t)$ at each step, we consider a baseline that uses the dense dynamics model $P_{\phi_c}$ without the causal mask $M$. We then prioritize causal information by maximizing the difference in empowerment gain between the causal and dense dynamics models.

We first denote the empowerment gain of the causal dynamics model and dense dynamics model as $\mathcal{E}_{\phi_c}(s|M) = \max_a \mathcal{I}(s_{t+1}; a_t \mid s_t; \phi_c, M)$ and $\mathcal{E}_{\phi_c}(s) = \max_a \mathcal{I}(s_{t+1}; a_t \mid s_t; \phi_c)$, respectively. Here, $\mathcal{E}_{\phi_c}(s)$ corresponds to the dynamics model without considering causal structures.

Then, we have the following learning objective:

$$\max_{a \sim \pi_e(a|s)} \mathbb{E}_{(s,a,s') \sim \mathcal{D}} \left[ \mathcal{E}_{\phi_c}(s|M) - \mathcal{E}_{\phi_c}(s) \right]. \tag{7}$$

In practice, we employ the estimated $\hat{\mathcal{E}}_{\phi_c}(s \mid M)$ and $\hat{\mathcal{E}}_{\phi_c}(s)$ with the policy $\pi_e$ for computing, specifically:

$$\hat{\mathcal{E}}_{\phi_c}(s_t|M) = \max_{a \sim \pi_e(a|s)} \mathbb{E}_{\pi_e(a_t|s_t) p_{\phi_c}(s_{t+1}|s_t, a_t, M)} \left[ \log P_{\phi_c}(s_{t+1} \mid s_t, a_t; M, \phi_c) - \log P(s_{t+1}|s_t) \right], \tag{8}$$

and:

$$\hat{\mathcal{E}}_{\phi_c}(s_t) = \max_{a \sim \pi_e(a|s)} \mathbb{E}_{\pi_e(a_t|s_t) p_{\phi_c}(s_{t+1}|s_t, a_t)} \left[ \log P_{\phi_c}(s_{t+1} \mid s_t, a_t; \phi_c) - \log P(s_{t+1}|s_t) \right], \tag{9}$$

where $P(s_{t+1}|s_t)$ is the conditional distribution of the current state. Hence, the objective function Eq. 7 is derived as:

$$\max_{a \sim \pi_e(a|s)} \mathcal{H}(s_{t+1} \mid s_t; M) - \mathcal{H}(s_{t+1} \mid s_t) + \mathbb{E}_{a \sim \pi_e(a|s)} \left[ \mathbb{KL} \left( P_{\phi_c}(s_{t+1} \mid s_t, a_t; M) \| P_{\phi_c}(s_{t+1} \mid s_t, a_t) \right) \right], \tag{10}$$

where $\mathcal{H}(s_{t+1} \mid s_t; M)$ and $\mathcal{H}(s_{t+1} \mid s_t)$ denote the entropy at time $t + 1$ under the causal dynamics model and dense dynamics model, respectively. For simplicity, we update $\pi_e$ by optimizing the KL term.

**Model Optimization**   In Step 2, we fix the dynamics model $P_{\phi_c}$ and further fine-tune the causal mask $M$ and the reward model $P_{\varphi_r}$. We adopt an alternating optimization with the policy $\pi_e$ to optimize the causal mask. Specifically, given $M$, we first optimize $\pi_e$. The policy $\pi_e$ is designed to collect controllable trajectories by maximizing the distance of empowerment between causal and dense models. These collected trajectories are then used to optimize both the causal structure $M$ and reward model $P_{\varphi_r}$.

### 3.3 STEP 3: POLICY LEARNING WITH CURIOSITY REWARD

We learn the downstream task policy based on the optimized causal structure. To mitigate potential overfitting of the causality learned in Steps 1&2, we incorporate a curiosity-based reward as an intrinsic motivation objective or exploration bonus, in conjunction with a task-specific reward, to prevent overfitting during task learning:

$$
r_{\text{cur}}(s,a) = \mathbb{E}_{(s_t,a_t,s_{t+1})\sim\mathcal{D}}\left[ \mathbb{KL}\Big( P_{\text{env}}(s_{t+1}|s_t,a_t) \,\Big\|\, P_{\phi_c,M}(s_{t+1}|s_t,a_t;\phi_c,M)\Big) \right.
$$
$$
\left. -\mathbb{KL}\Big( P_{\text{env}}(s_{t+1}|s_t,a_t) \,\Big\|\, P_{\phi_c}(s_{t+1}|s_t,a_t;\phi_c)\Big) \right] \tag{11}
$$

where $P_{\text{env}}$ is the ground truth dynamics collected from the environment. By taking account of $r_{\text{cur}}$, we encourage the agent to explore states that the causal dynamics cannot capture but the dense dynamics can from the true environment dynamics, thus preventing the policy from being overly conservative due to model learning with trajectories. Hence, the shaped reward function is:

$$
r(s,a) = r_{\text{task}}(s,a) + \lambda r_{\text{cur}}(s,a), \tag{12}
$$

where $r_{\text{task}}(s,a)$ is the task reward, $\lambda$ is a balancing hyperparameter. In section D.8, we conduct ablation experiments to thoroughly analyze the impact of different shaped rewards, including curiosity, causality and original task rewards.

## 4 PRACTICAL IMPLEMENTATION

We introduce the practical implementation of `ECL` for casual dynamics learning with empowerment-driven exploration and task learning. The proposed framework for the entire learning process is illustrated in Figure 2, comprising three steps and the full pipeline is listed in Algorithm 1.

**Step 1: Model Learning** Initially, following (Wang et al., 2022c), we use a transition collection policy $\pi_{\text{collect}}$ by formulating a reward function that incentivizes selecting transitions that cover more state-action pairs to expose causal relationships thoroughly. We train the dynamics model $P_{\phi_c}$ by maximizing the log-likelihood $\mathcal{L}_{\text{dyn}}$, following Eq. 4. Then, we employ the causal discovery approach for learning causal mask $M$ by maximizing the log-likelihood $\mathcal{L}_{\text{c-dyn}}$ followed Eq. 5. Subsequently, we train the reward model $P_{\varphi_{\text{r}}}$ with the state abstraction $\phi_c(s \mid M)$ by maximizing the likelihood.

**Step 2: Model Optimization** We execute the empowerment-driven exploration by $\max_{a\sim\pi_e(a|s)} \mathbb{E}_{s_t,a_t,s_{t+1}\sim\mathcal{D}}\left[\mathcal{E}_{\phi_c}(s|M) - \mathcal{E}_{\phi_c}(s)\right]$ followed Eq. 7 with causal dynamics model and dense dynamics model for policy $\pi_e$ learning. Furthermore, the learned policy $\pi_e$ is used to sample transitions for updating casual mask $M$ and reward model. We alternately perform empowerment-driven exploration for policy learning and causal model optimization.

**Step 3: Policy Learning** During downstream task learning, we incorporate the causal effects of different actions as curiosity rewards combined with the task reward, following Eq. 12. We maximize the discounted cumulative reward to learn the policy by the cross entropy method (CEM) (Rubinstein, 1997). Specifically, The causal model is used to execute dynamic state transitions defined in Eq. 2. The reward model evaluates these transitions and provides feedback in the form of rewards. The CEM handles the planning process by leveraging the predictions from the causal and reward models to optimize the task's objectives effectively.

## 5 EXPERIMENTS

We aim to answer the following questions in experimental evaluation: (i) How does the performance of `ECL` compare to other causal and dense models across different environments for tasks and dynamics learning, including pixel-based tasks? (ii) Does `ECL` improve causal discovery by eliminating more irrelevant state dimensions interference, thereby enhancing learning efficiency and generalization towards the empowerment gain? (iii) Whether different causal discovery methods in step 1 and 2, impact policy performance? What are the effects when combine the step 1 and 2? (iv) What are the effects of the components and hyperparameters in `ECL`?

### 5.1 SETUP

**Environments.** We select 3 different environments for basic experimental evaluation. **Chemical** (Ke et al., 2021): The task is to discover the causal relationship (Chain, Collider & Full) of chemical

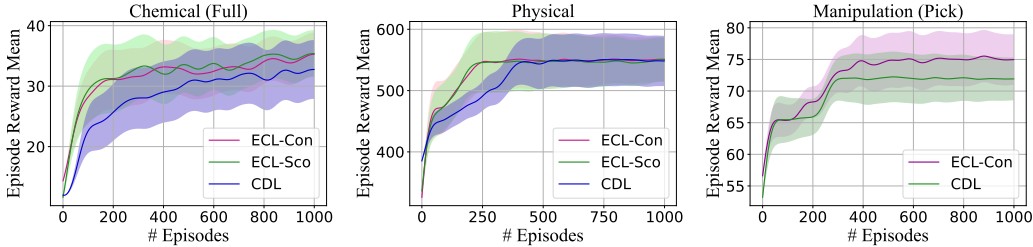

Figure 3: The task learning of episodic reward in three environments of **ECL-Con** (**ECL-C**) and **ECL-Sco** (**ECL-S**).

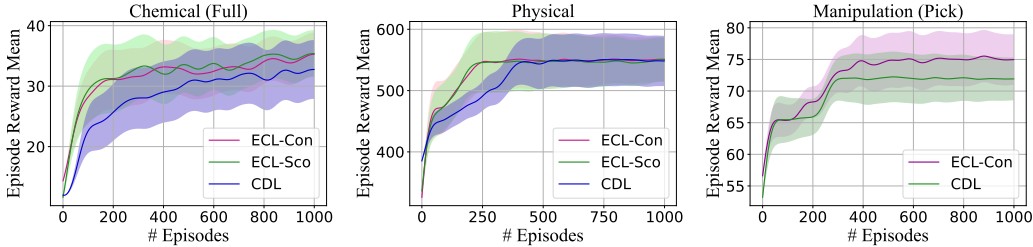

Figure 4: The learning curves of episodic reward in three different environments and the shadow is the standard error.

items which proves the learned dynamics and explains the behavior without spurious correlations. **Manipulation** (Wang et al., 2022c): The task is to prove dynamics and policy for difficult settings with spurious correlations and multi-dimension action causal influence. **Physical** (Ke et al., 2021): a dense mode Physical environment. Furthermore, we also include 3 pixel-based environments of **Modified Cartpole** (Liu et al., 2024), Robedesk (Wang et al., 2022a) and **Deep Mind Control** (**DMC**) (Wang et al., 2022a) for evaluation in latent state environments. For the details of the environment setup, please refer to Appendix D.2.

**Baselines.** We compare **ECL** with 4 causal and 2 standard MBRL methods. **CDL** (Wang et al., 2022c): infers causal relationships between the variables for dynamics learning with Conditional Independence Test (CIT) of constraint-based causal discovery. **REG** (Wang et al., 2021): Action-sufficient state representation based on regularization of score-based causal discovery. **GRADER** (Ding et al., 2022): generalizing goal-conditioned RL with CIT by variational causal reasoning. **IFactor** (Liu et al., 2024): a causal framework to model four distinct categories of latent state variables within the RL system for pixel-based environments. **GNN** (Ke et al., 2021): a graph neural network with dense dependence for each state variable. **Monolithic** (Wang et al., 2022c): a Multi-Layer Perceptron (MLP) network that takes all state variables and actions for prediction. For **ECL**, we employ both conditional independence testing (constraint-based (**ECL-Con**)) used in (Wang et al., 2022c) and mask learning by sparse regularization (score-based (**ECL-Sco**)) used in (Huang et al., 2022). We also combine IFactor (Liu et al., 2024) for pixel-based tasks learning detailed in Appendix D.2.2.

**Evaluation Metrics.** In tasks learning, we utilize episodic reward and task success as evaluation criteria for downstream tasks. For causal dynamics learning, we employ five metrics to evaluate the learned causal graph and assess the mean accuracy for dynamics predictions of future states in both ID and OOD. For pixel-based tasks, we use average return and visualization results for evaluation[1].

## 5.2 RESULTS

### 5.2.1 TASK LEARNING

We evaluate each method with the following 7 downstream tasks in the chemical (C), physical (P) and the manipulation (M) environments. **Match** (C): match the object colors with goal colors individually. **Push** (P): use the heavier object to push the lighter object to the goal position. **Reach** (M): move the end-effector to the goal position. **Pick** (M): pick the movable object to the goal position. **Stack** (M): stack the movable object on the top of the unmovable object.

---

[1]We conduct each experiment using 4 random seeds.

As shown in Fig. 3, compared to dense dynamics models GNN and MLP, as well as the causal approaches CDL and REG, **ECL-Con** attains the highest reward across 3 environments. Notably, **ECL-Con** outperforms other methods in the intricate manipulation tasks. Furthermore, **ECL-Sco** surpasses REG, elevating model performance and achieving a reward comparable to CDL. The proposed curiosity reward encourages exploration and avoids local optimality during the policy learning process. For full results, please refer to Appendix D.5.

Additionally, Figure 4 depicts the learning curves across three environments. Across these diverse settings, **ECL** exhibits elevated sample efficiency compared to CDL and higher reward attainment. The introduction of curiosity reward bonus enables efficient exploration of strategies, thus averting the risk of falling into local optima. Overall, our proposed intrinsic-motivated causal empowerment learning framework demonstrates improved stability and learning efficiency. We also evaluate the effect of combining steps 1 and 2, as shown in Appendix D.8. For full experimental results in property analysis and ablation studies, please refer to Appendix D.7 and D.8.

**Sample Efficiency Analysis.** After validating the effectiveness of **ECL** in reward learning, we further substantiate the improvements in sample efficiency of **ECL** for task execution. As depicted in Figure 5, we illustrate task success in both collider and manipulation reach tasks. The compared experimental results underscore the efficiency of our approach, demonstrating enhanced sample efficiency across different environments.

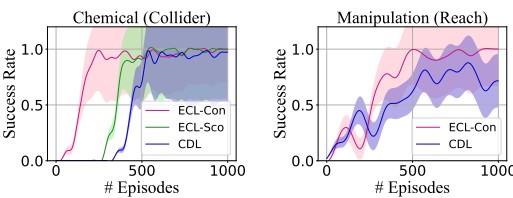

Figure 5: Success rate in collider and manipulation environments and the shadow is the standard error.

### 5.2.2 CAUSAL DYNAMICS LEARNING

**Causal Graph Learning.** To evaluate the efficacy of our proposed method for learning causal relationships, we first conduct experimental analyses across three chemical environments, employing five evaluation metrics. We conduct causal learning based on the causal discovery with Con and Sco respectively. The comparative results using the same causal discovery methods are presented in Table 1, with each cell containing the comparative results for that method across different scenarios. These results demonstrate the superior performance of our approach in causal reasoning, exhibiting both effectiveness and robustness as evinced by the evaluation metrics of F1 score and ROC AUC (Wang et al., 2022c). All results exceed 0.90. Notably, our approach exhibits exceptional learning capabilities in chemical chain and collider environments. Moreover, it significantly enhances models performance when handling more complex full causal relationships, underscoring its remarkable capability in grasping intricate causal structures. This proposed causal empowerment framework facilitates more precise uncovering of causal relationships by actively using the causal structure.

**Visualization.** Moreover, we visually compare the inferred causal graph with the ground truth graph in terms of edge accuracy. The results depicted in Figure 6 illustrate the causal graphs of **ECL-Sco** compared to REG and GRADER in the collider environment. For nodes exhibiting strong causality, **ECL-Sco** achieves fully accurate learning and substantial accuracy enhancements compared to REG. Concurrently, **ECL-Sco** elucidates the causality between action and state more effectively. Furthermore, **ECL-Sco** mitigates interference from irrelevant causal nodes more proficiently than GRADER. The causal graph learned in the complex manipulation environment shown in Figure 15, demonstrates that **ECL** effectively excludes irrelevant state dimensions to avoid the influence of spurious correlations. These findings substantiate that the proposed method attains superior performance compared to other causal discovery methods in causal learning.

**Predicting Future States.** Given the current state and a sequence of actions, we evaluate the accuracy of each method's prediction, for states both ID and OOD. We evaluate each method for one step prediction on 5K transitions, for both ID and OOD states. To create OOD states, we change object positions in the chemical environment and marker positions in the manipulation environment to unseen values, followed (Wang et al., 2022c).

Figure 7 illustrates the prediction results across four environments. In the ID settings, our proposed methods, based on both Sco and Con, achieve performance on par with GNNs and MLPs, while

Table 1: Experimental results on causal graph learning in three chemical environments.

| Metrics | Methods | Chain | Collider | Full |
|---|---|---|---|---|
| **Accuracy** | **ECL**/CDL | 1.00±0.00/1.00±0.00 | 1.00±0.00/1.00±0.00 | **1.00±0.00**/0.99±0.00 |
| | **ECL**/REG | 0.99±0.00/0.99±0.00 | 0.99±0.00/0.99±0.00 | **0.99±0.01**/0.98±0.00 |
| **Recall** | **ECL**/CDL | **1.00±0.00**/0.99±0.01 | 1.00±0.00/1.00±0.00 | **0.97±0.01**/0.92±0.02 |
| | **ECL**/REG | **1.00±0.00**/0.94±0.01 | **0.99±0.01**/0.89±0.09 | **0.90±0.02**/0.79±0.01 |
| **Precision** | **ECL**/CDL | 1.00±0.00/1.00±0.00 | 1.00±0.00/1.00±0.00 | 0.96±0.02/ **0.97±0.02** |
| | **ECL**/REG | 0.99±0.01/0.99±0.01 | 0.99±0.01/0.99±0.01 | **0.97±0.03**/0.92±0.05 |
| **F1 Score** | **ECL**/CDL | **1.00±0.00**/0.99±0.01 | 1.00±0.00/1.00±0.00 | **0.97±0.01**/0.94±0.01 |
| | **ECL**/REG | **0.99±0.00**/0.96±0.01 | **0.99±0.00**/0.94±0.05 | **0.93±0.02**/0.85±0.02 |
| **ROC AUC** | **ECL**/CDL | **1.00±0.00**/0.99±0.01 | 1.00±0.00/1.00±0.00 | **0.98±0.01**/0.96±0.01 |
| | **ECL**/REG | 0.99±0.01/0.99±0.01 | **0.99±0.01**/0.93±0.04 | 0.95±0.01/0.95±0.01 |

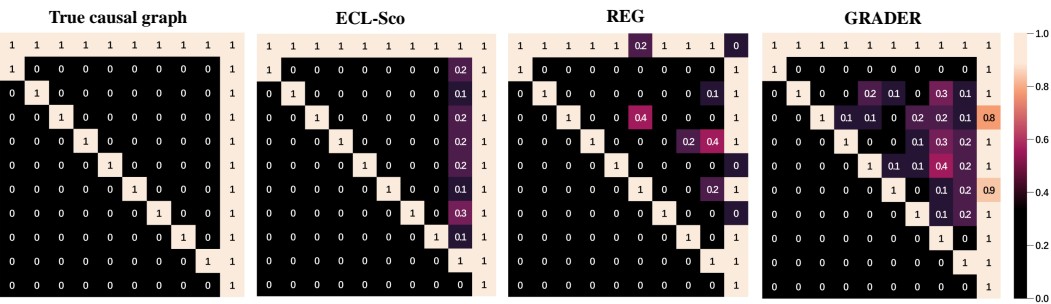

Figure 6: The causal graph comparison in the chemical collider environment.

significantly elevating performance in the intricate manipulation environment. These findings validate the efficacy of our proposed approach for causal learning. For the OOD settings, our method attains comparable performance to the ID setting. These results demonstrate strong generalization and robustness capabilities compared to GNNs and MLPs. Moreover, it outperforms CDL and REG. The comprehensive experimental results substantiate the proficiency of our proposed method in accurately uncovering causal relationships and enhancing generalization abilities. For full results of causal dynamics learning, please refer to Appendix D.3 and D.4.

### 5.2.3 Pixel-Based Task Learning

In complex pixel-based robodesk task, where video backgrounds serve as distractors, **ECL** effectively learns controllable policies for changing background colors to green, as shown in Figure 8. Additionally, **ECL** surpasses IFactor in terms of average return. These results further validate **ECL**'s efficacy in pixel-based tasks and its ability to overcome spurious correlations (video backgrounds). For more results in pixel-based tasks, please refer to Appendix D.6.

## 6 Related Work

**Causal MBRL** MBRL involves training a dynamics model by maximizing the likelihood of collected transitions, known as the world model (Moerland et al., 2023; Janner et al., 2019; Nguyen et al., 2021; Zhao et al., 2021). Due to the exclusion of irrelevant factors from the environment through state abstraction, the application of causal inference in MBRL can effectively improve sample efficiency and generalization (Ke et al., 2021; Mutti et al., 2023b; Hwang et al., 2023). Wang et al. (2021) propose a constraint-based causal dynamics learning that explicitly learns causal dependencies by action-sufficient state representations. GRADER (Ding et al., 2022) executes variational inference by regarding the causal graph as a latent variable. CDL (Wang et al., 2022c) is a causal dynamics learning method based on CIT. CDL employs conditional mutual information to compute the causal relationships between different dimensions of states and actions. For additional related work, please refer to Appendix B.

**Empowerment in RL** Empowerment is an intrinsic motivation to improve the controllability over the environment (Klyubin et al., 2005; Salge et al., 2014). This concept is from the information-theoretic framework, wherein actions and future states are viewed as channels for information

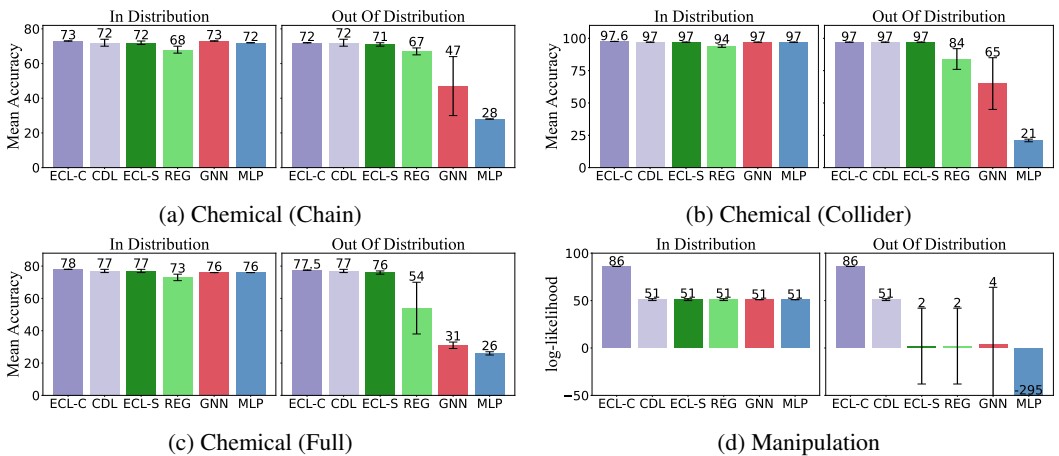

Figure 7: Prediction performance (%) on ID and OOD states of `ECL-Con` (`ECL-C`) and `ECL-Sco` (`ECL-S`). The mean score is marked on the top of each bar.

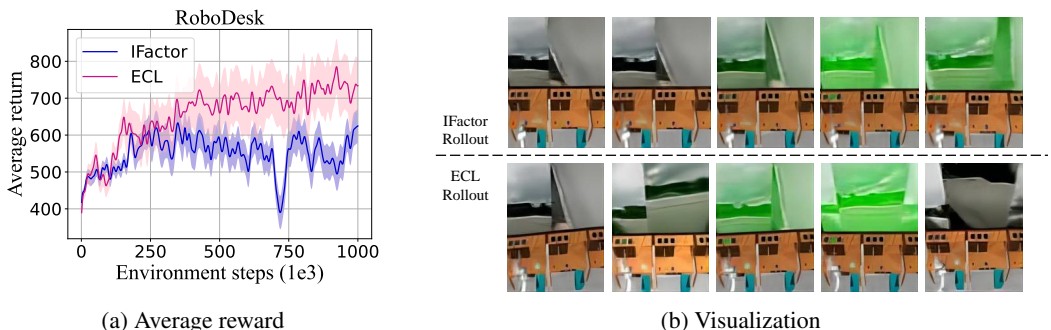

Figure 8: The compared results with IFactor and visualized trajectories in Robodesk environment.

transmission. In RL, empowerment is applied to uncover more controllable associations between states and actions or skills (Mohamed & Jimenez Rezende, 2015; Bharadhwaj et al., 2022; Choi et al., 2021; Eysenbach et al., 2018). By quantifying the influence of different behaviors on state transitions, empowerment encourages the agent to explore further to enhance its controllability over the system (Leibfried et al., 2019; Seitzer et al., 2021). Maximizing empowerment $\max_\pi I$ can be used as the learning objective, empowering agents to demonstrate intelligent behavior without requiring predefined external goals.

## 7 CONCLUSION

This paper proposes a method-agnostic framework of empowerment through causal structure learning in MBRL to improve controllability and learning efficiency by iterative policy learning and causal structure optimization. We maximize empowerment under causal structure to prioritize controllable information and optimize causal dynamics and reward models to guide downstream task learning. Extensive experiments across 6 environments included pixel-based tasks substantiate the remarkable performance of the proposed framework.

**Limitation and Future Work** `ECL` implicitly enhances the controllability but does not explicitly tease apart different behavioral dimensions. In our future work, we plan to extend this framework in several directions. First, we aim to disentangle behaviors and explore entropy relaxation methods to enhance empowerment, particularly for real-world robotics tasks (Collaboration et al., 2023). Second, while the current framework does not account for changing dynamics, we intend to incorporate insights from recent advancements in local causal discovery (Hwang et al., 2023) and modeling non-stationary change factors (Huang et al., 2020) to enhance the causal discovery component. Third, we plan to leverage pre-trained 3D or object-centric visual dynamics models (Shi et al., 2024; Wang et al., 2023b; Luo et al., 2024; Team et al., 2024) to scale our approach to real-world robotics applications. These directions will be pursued in future work.

## REPRODUCIBILITY STATEMENT

We provide the source code of `ECL` in the supplementary material. The implementation details of experimental settings and platforms are shown in Appendix D.

## ACKNOWLEDGMENT

This work was supported in part by the National Natural Science Foundation of China under Grant 62276128, Grant 62192783; in part by the Jiangsu Science and Technology Major Project BG2024031; in part by the Natural Science Foundation of Jiangsu Province under Grant BK20243051; the Fundamental Research Funds for the Central Universities(14380128); in part by the Collaborative Innovation Center of Novel Software Technology and Industrialization.

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

Supplementary material
Towards Empowerment Gain through
Causal Structure Learning in
Model-Based Reinforcement Learning

## Table of Contents

# A   DISCUSSIONS AND BROADER IMPACT

## A.1   DISCUSSIONS

From the technical perspective, our work explores leveraging causal structure to enhance empowerment for efficient policy learning, enabling better control of the environment in MBRL. We propose a framework that can effectively combine diverse causal discovery methods. This holistic approach not only refines policy learning but also ensures that the causal model remains adaptable and accurate, even when faced with novel or shifting environmental conditions. **ECL** demonstrates improved learning efficiency and generalization compared to other causal MBRL methods across six different RL environments, including pixel-based tasks. Simultaneously, **ECL** achieves more accurate causal relationship discovery, overcoming spurious correlation present in the environment. While **ECL** demonstrated strengths in the accurate causal discovery and overcoming spurious correlation, disentangling controllable behavioral dimensions remains a limitation. Our implicit empowerment approach enhances the policy's control over the environment but does not explicitly tease apart different behavioral axes. Explicitly disentangling controllable behavioral dimensions could be an important future work to further improve behavioral control and empowerment. Additionally, our current approach involves substantial data collection and model optimization efforts, which can hinder training efficiency. Moving forward, we aim to further streamline our framework to enable more efficient policy training and causal structure learning. Enhancing computational performance while maintaining accuracy will be a key focus area for future iterations of this work. In the empowerment maximization described by Eq. 10, we currently omit two entropy terms. In our future work, we plan to explore additional entropy relaxation methods to further optimize this causal empowerment learning objective.

## A.2   BROADER IMPACT

We believe that our work contributes to developing a more interpretable model via causality for model-based reinforcement learning (MBRL), which could benefit various domains, including robotics (Ibarz et al., 2021; Lee et al., 2021), healthcare (Yu et al., 2021; Tang et al., 2022), and scientific discovery (Runge et al., 2019; Nowack et al., 2020; Castro et al., 2020; Liu et al., 2022; Wang et al., 2023a), particularly in contexts involving causal modeling and decision-making agents. While RL-based approaches may present potential risks in these domains, we do not identify any risks uniquely introduced by this work that require specific discussion.

# B   ADDITIONAL RELATED WORKS

## B.1   MODEL-BASED REINFORCEMENT LEARNING

MBRL involves training a dynamics model by maximizing the likelihood of collected transitions, known as the world model, as well as learning a reward model (Moerland et al., 2023; Janner et al., 2019; Hao et al., 2023; Cao et al., 2022; Yang & Gao, 2020). Based on learned models, MBRL can execute downstream task planning (Nguyen et al., 2021; Zhao et al., 2021), data augmentation (Pitis et al., 2022; Okada & Taniguchi, 2021; Yu et al., 2020), and Q-value estimation (Wang et al., 2022b; Amos et al., 2021). MBRL can easily leverage prior knowledge of dynamics, making it more effective at enhancing policy stability and generalization. However, when faced with high-dimensional state spaces and confounders in complex environments, the dense models learned by MBRL suffer from spurious correlations and poor generalization (Wang et al., 2022c; Bharadhwaj et al., 2022). To tackle these issues, causal inference approaches are applied to MBRL for state abstraction, removing unrelated components (Hwang et al., 2023; Ding et al., 2022; Wang et al., 2024).

## B.2   CAUSALITY IN MBRL

Due to the exclusion of irrelevant factors from the environment through causality, the application of causal inference in MBRL can effectively improve sample efficiency and generalization (Ke et al., 2021; Mutti et al., 2023b; Liu et al., 2024; Urpí et al., 2024). Wang (Wang et al., 2021) proposes a regularization-based causal dynamics learning method that explicitly learns causal dependencies by regularizing the number of variables used when predicting each state variable. GRADER (Ding et al.,

2022) execute variational inference by regarding the causal graph as a latent variable. IFactor (Liu et al., 2024) is a general framework to model four distinct categories of latent state variables, capturing various aspects of information. CDL (Wang et al., 2022c) is a causal dynamics learning method based on conditional independence testing. CDL employs conditional mutual information to compute the causal relationships between different dimensions of states and actions, thereby explicitly removing unrelated components. However, it is challenging to strike a balance between explicit causal discovery and prediction performance, and the learned policy has lower controllability over the system. In this work, we aim to actively leverage learned causal structures to achieve effective exploration of the environment through empowerment, thereby learning controllable policies that generate data to further optimize causal structures.

## C NOTATIONS, ASSUMPTIONS AND PROPOSITIONS

### C.1 NOTATIONS

| Symbol | Description | Details |
|---|---|---|
| $s_t^i$ | $i$-th state at time $t$ | – |
| $a_t$ | action at time $t$ | – |
| $r_t$ | reward at time $t$ | – |
| $M^{s \to s}$ | Causal masks between actions and states | Trainable parameters in Eq. 5 |
| $M^{a \to s}$ | Causal masks between actions and states | Trainable parameters in Eq. 5 |
| $f$ | Dynamics function | Dynamics function of the MDPs |
| $R$ | Reward function | Reward function of the MDPs |
| $\mathcal{I}$ | Mutual information | – |
| $\mathcal{E}$ | Empowerment gain | – |
| $\phi_c$ | Parameters of dynamics model | Trainable parameters in Eq. 4 |
| $\varphi_r$ | Parameters of reward model | Trainable parameters in Eq. 6 |
| $\pi_e$ | Parameters of empowerment-driven policy | Trainable parameters in Eq. 7 |
| $\pi$ | Parameters of task policy | Trainable parameters for task policy learning |

Table 2: Notations used throughout the paper.

### C.2 DETAILED OBJECTIVE FUNCTIONS

To better illustrate the trainable parameters in each objective function, we mark the trainable ones in red as follows.

$$\mathcal{L}_{\text{dyn}} = \mathbb{E}_{(s_t, a_t, s_{t+1}) \sim \mathcal{D}} \left[ \sum_{i=1}^{d_S} \log P_{\phi_c}(s_{t+1}^i | s_t, a_t; \phi_c) \right] \tag{13}$$

$$\mathcal{L}_{\text{c-dyn}} = \mathbb{E}_{(s_t, a_t, s_{t+1}) \sim \mathcal{D}} \left[ \sum_{i=1}^{d_S} \log P_{\phi_c}(s_{t+1}^i | M^{s \to s^j} \odot s_t, M^{a \to s^j} \odot a_t; \phi_c) + \mathcal{L}_{\text{causal}} \right] \tag{14}$$

$$\mathcal{L}_{\text{rew}} = \mathbb{E}_{(s_t, a_t, r_t) \sim \mathcal{D}} \left[ \log P_{\varphi_r}(r_t | \phi_c(s_t \mid M), a_t) \right] \tag{15}$$

$$\max_{a \sim \pi_e(a|s)} \mathbb{E}_{(s_t, a_t, s_{t+1}) \sim \mathcal{D}} \left[ \mathcal{E}_{\phi_c}(s|M) - \mathcal{E}_{\phi_c}(s) \right]. \tag{16}$$

### C.3 ASSUMPTIONS AND PROPOSITIONS

**Assumption 1** *(d-separation (Pearl, 2009)) d-separation is a graphical criterion used to determine, from a given causal graph, if a set of variables X is conditionally independent of another set Y, given*

*a third set of variables Z. In a directed acyclic graph (DAG) $\mathcal{G}$, a path between nodes $n_1$ and $n_m$ is said to be blocked by a set S if there exists a node $n_k$, for $k = 2, \cdots, m-1$, that satisfies one of the following two conditions:*

*(i) $n_k \in S$, and the path between $n_{k-1}$ and $n_{k+1}$ forms $(n_{k-1} \rightarrow n_k \rightarrow n_{k+1})$, $(n_{k-1} \leftarrow n_k \leftarrow n_{k+1})$, or $(n_{k-1} \leftarrow n_k \rightarrow n_{k+1})$.*

*(ii) Neither $n_k$ nor any of its descendants is in S, and the path between $n_{k-1}$ and $n_{k+1}$ forms $(n_{k-1} \rightarrow n_k \leftarrow n_{k+1})$.*

*In a DAG, we say that two nodes $n_a$ and $n_b$ are d-separated by a third node $n_c$ if every path between nodes $n_a$ and $n_b$ is blocked by $n_c$, denoted as $n_a \perp\!\!\!\perp n_b | n_c$.*

**Assumption 2** *(Global Markov Condition (Spirtes et al., 2001; Pearl, 2009)) The state is fully observable and the dynamics is Markovian. The distribution p over a set of variables $\mathcal{V} = (s_t^1, \cdots, s_t^d, a_t^1, \cdots, a_t^d, r_t)^T$ satisfies the global Markov condition on the graph if for any partition $(\mathcal{S}, \mathcal{A}, \mathcal{R})$ in $\mathcal{V}$ such that if $\mathcal{A}$ d-separates $\mathcal{S}$ from $\mathcal{R}$, then $p(\mathcal{S}, \mathcal{R}|\mathcal{A}) = p(\mathcal{S}|\mathcal{A}) \cdot p(\mathcal{R}|\mathcal{A})$*

**Assumption 3** *(Faithfulness Assumption (Spirtes et al., 2001; Pearl, 2009)) For a set of variables $\mathcal{V} = (s_t^1, \cdots, s_t^d, a_t^1, \cdots, a_t^d, r_t)^T$, there are no independencies between variables that are not implied by the Markovian Condition.*

**Assumption 4** *Under the assumptions that the causal graph is Markov and faithful to the observations, the edge $s_t^i \rightarrow s_{t+1}^i$ exists for all state variables $s^i$.*

**Assumption 5** *No simultaneous or backward edges in time.*

**Theorem 1** *Based on above 5 assumptions, we define the conditioning set $\{a_t, s_t \setminus s_t^i\} = \{a_t, s_t^1, \ldots s_t^{i-1}, s_t^{i+1}, \ldots\}$. If $s_t^i \not\perp\!\!\!\perp s_{t+1}^j | \{a_t, s_t \setminus s_t^i\}$, then $s_t^i \rightarrow s_{t+1}^j$. Similarly, if $a_t^i \not\perp\!\!\!\perp s_{t+1}^j | \{a_t \setminus a_t^i, s_t\}$, then $a_t^i \rightarrow s_{t+1}^j$.*

**Proposition 1** *Under the assumptions that the causal graph is Markov and faithful to the observations, there exists an edge from $a_t^i \rightarrow s_{t+1}^j$ if and only if $a_t^i \not\perp\!\!\!\perp s_{t+1}^j | \{a_t \setminus a_t^i, s_t\}$, then $a_t^i \rightarrow s_{t+1}^j$.*

*Proof.* We first prove that if there exists an edge from $a_t^i$ to $s_{t+1}^j$, then $a_t^i \not\perp\!\!\!\perp s_{t+1}^j | \{a_t \setminus a_t^i, s_t\}$. We prove it by contradiction. Suppose that $a_t^i$ is independent of $s_{t+1}^j$ given $\{a_t \setminus a_t^i, s_t\}$. According to the faithfulness assumption, we can infer this independence from the graph structure. If $a_t^i$ is independent of $s_{t+1}^j$ given $\{a_t \setminus a_t^i, s_t\}$, then there cannot be a directed path from $a_t^i$ to $s_{t+1}^j$ in the graph. Hence, there is no edge between $a_t^i$ and $s_{t+1}^j$. This contradicts our initial statement about the existence of this edge.

Now, we prove the converse: if $a_t^i \not\perp\!\!\!\perp s_{t+1}^j | \{a_t \setminus a_t^i, s_t\}$, then there exists an edge from $a_t^i$ to $s_{t+1}^j$. Again, we use proof by contradiction. Suppose there is no edge between $a_t^i$ and $s_{t+1}^j$ in the graph. Due to the Markov assumption, the lack of an edge between these variables implies their conditional independence given $\{a_t \setminus a_t^i, s_t\}$. This contradicts our initial statement that $a_t^i \not\perp\!\!\!\perp s_{t+1}^j | \{a_t \setminus a_t^i, s_t\}$. Therefore, there must exist an edge from $a_t^i$ to $s_{t+1}^j$.

**Proposition 2** *Under the assumptions that the causal graph is Markov and faithful to the observations, there exists an edge from $s_t^i \rightarrow s_{t+1}^j$ if and only if $s_t^i \not\perp\!\!\!\perp s_{t+1}^j | \{a_t, s_t \setminus s_t^i\}$.*

The proof of Proposition 2 follows a similar line of reasoning to that of Proposition 1. Consequently, the two propositions collectively serve as the foundation for deriving Theorem 1.

# D DETAILS ON EXPERIMENTAL DESIGN AND RESULTS

## D.1 EXPERIMENTAL ENVIRONMENTS

We select three different types environments for basic experimental evaluation, as shown in Figure 9.

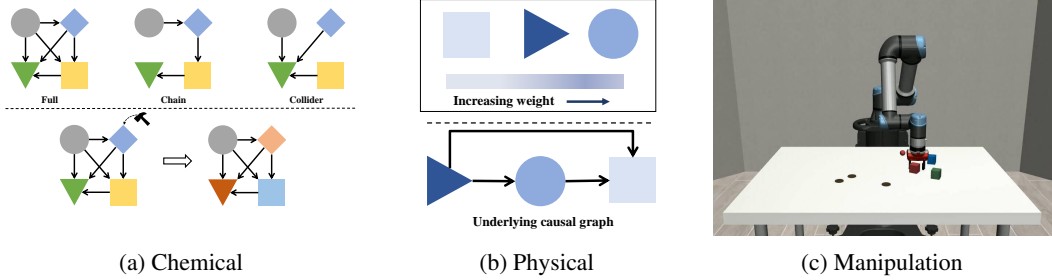

(a) Chemical        (b) Physical        (c) Manipulation

Figure 9: Three basic experimental environments.

**Chemical** In chemical environment, we aim to discover the causal relationship (Chain, Collider & Full) of chemical items which will prove the learned dynamics and explain the behavior without spurious correlations. Meanwhile, in the downstream tasks, we evaluate the proposed methods by episodic reward and success rate. The reward function is defined as follows:

**Match:** match the object colors with goal colors individually:

$$r^{\text{match}} = \sum_{i=1}^{10} \mathbb{1}[m_t^i = g^i] \tag{17}$$

where $\mathbb{1}$ is the indicator function, $m_t^i$ is the current color of the $i$-object, and $g^i$ is the goal color of the $i$-object.

**Manipulation** In the manipulation environment, we aim to prove the learned dynamics and policy for difficult settings with spurious correlations and multi-dimension action causal influence. The state space consists of the robot end-effector (EEF) location ($\mathbb{R}^3$), gripper (grp) joint angles ($\mathbb{R}^2$), and locations of objects and markers ($6 \times \mathbb{R}^3$). The action space includes EEF location displacement ($\mathbb{R}^3$) and the degree to which the gripper is opened ($[0, 1]$). In each episode, the objects and markers are reset to randomly sampled poses on the table. The task reward functions of **Reach**, **Pick** and **Stack** are followed (Wang et al., 2022c).

**Physical** In addition to the chemical and manipulation environment, we also evaluate our method in the physical environment. In a $5 \times 5$ grid-world, there are 5 objects and each of them has a unique weight. The state space is 10-dimensional, consisting of x, y positions (a categorical variable over 5 possible values) of all objects. At each step, the action selects one object, moves it in one of 4 directions or lets it stay at the same position (a categorical variable over 25 possible actions). During the movement, only the heavier object can push the lighter object (the object won't move if it tries to push an object heavier than itself). Meanwhile, the object cannot move out of the grid-world nor can it push other lighter objects out of the grid-world. Moreover, the object cannot push two objects together, even when both of them are lighter than itself (Dense model mode). The task reward function is defined as follows:

**Push:** calculate the average distance between the current node and the target location:

$$r^{\text{match}} = \frac{1}{5} \sum_{i=1}^{5} \text{dis}(o_i, t_i) \tag{18}$$

where $\text{dis}(\cdot)$ is the distance between two objects position. $o_i$ is the position of current node and $t_i$ is the position of target node.

### D.1.1 PIXEL-BASED ENVIRONMENTS

Importantly, to evaluate the performance of our proposed **ECL** framework in latent state environments, we select three distinct categories of pixel-based environments with distractors for assessment, as shown in Figure 10. We employ IFactor (Liu et al., 2024) as our baseline method and used its encoders to process visual inputs. Subsequently, we apply the proposed **ECL** framework for policy

learning. The parameter settings for these three environments are kept consistent with the default configurations of IFactor.

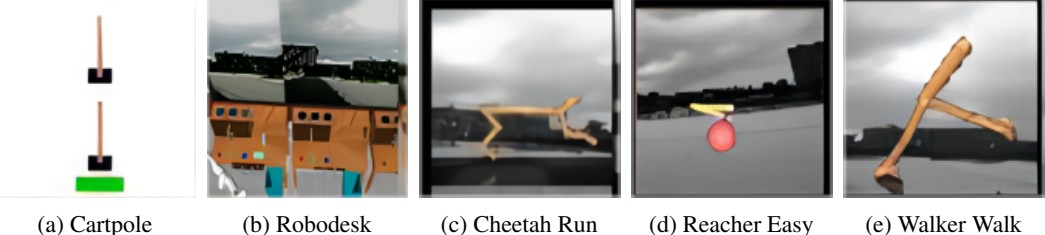

(a) Cartpole  (b) Robodesk  (c) Cheetah Run  (d) Reacher Easy  (e) Walker Walk

Figure 10: 3 pixel-based experimental environments with 5 tasks.

**Modified Cartpole**  We select a variant of the original Cartpole environment by incorporating two distractors (Liu et al., 2024), as shown in Figure 10(a). The first distractor is an uncontrollable Cartpole located in the upper portion of the image, which is irrelevant to the rewards. The second distractor is a controllable but reward-irrelevant green light positioned below the reward-relevant Cartpole in the lower part of the image.

**Robodesk**  We select a variant of Robodesk (Kannan et al., 2021; Wang, 2022), which includes realistic noise element with a dynamic video background, as shown in Figure 10(b). In this task, the objective for the agent is to change the hue of a TV screen to green using a button press, while ignoring the distractions from the video background.

**Deep Mind Control**  We also consider variants of DMC (Wang et al., 2022a; Tassa et al., 2018), where a dynamic video background is introduced to the original DMC environment as distractor. We select cheetah Run, reacher Easy and walker Walk three specific tasks for evaluation, as shown in Figure 10(c, d, e).

## D.2 EXPERIMENTAL SETUP

### D.2.1 DYNAMICS LEARNING IMPLEMENTATION DETAILS

We present the architectures of the proposed method across all environments in Table 3. For all activation functions, the Rectified Linear Unit (ReLU) is employed. Additionally, we summarize the hyperparameters for causal mask learning used in all environments for `ECL-Con` and `ECL-Sco` in Table 4. Regarding the other parameter settings, we adhered to the parameter configurations established in CDL (Wang et al., 2022c) and ASR (Huang et al., 2022). Moreover, The policy $\pi_{collect}$ is trained with a reward function $r = \tanh(\sum_{j=1}^{d_\mathcal{S}} \log \frac{p(s_{t+1}^j|s_t,a_t)}{p(s_{t+1}^j|\text{PA}_{sj})})$. This reward function measures the prediction difference between the dense predictor and the current causal predictor, following the approach described in CDL (Wang et al., 2022c).

chemical, manipulation, and physical environments, we utilize well-defined feature spaces for states and actions, which are explicitly designed for causal structure learning. For pixel-based environments such as DMC, Cartpole, and RoboDesk, ECL operates on latent states extracted by visual encoders. These encoders are supported by the identifiability theory proposed in IFactor (Liu et al., 2024), which ensures that these latent states can effectively map to the true states. While establishing identifiability is not the primary focus of our work, we leverage IFactor's encoders and include comparisons with IFactor in our experiments. Importantly, even in these settings, we can learn meaningful causal graphs.

### D.2.2 TASK LEARNING IMPLEMENTATION DETAILS

We list the downstream task learning architectures of the proposed method across all environments in Table 5. We outline the parameter configurations for the reward predictor, as well as the settings employed for the cross-entropy method that is applied. For pixel-based task learning, we leverage the

Table 3: Architecture settings in all environments.

| Architecture | Environments | | |
|---|---|---|---|
| | Chemical | Physical | Manipulation |
| feature dimension | 64 | 128 | 128 |
| predictive networks | [64,32] | [128,128] | [128,64] |
| training steps | 500K | 500K | 32M |
| max step of environment | 50 | 100 | 250 |
| batch size | | 64 | |
| learing rate | | 1e-4 | |
| max sample time | | 128 | |
| prediction step during training | | 2 | |

Table 4: Hyperparameters for causal mask learning in all environments.

| Method | hyperparameters | Environments | | |
|---|---|---|---|---|
| | | Chemical | Physical | Manipulation |
| ECL−Con | CMI threshold | 0.02 | 0.01 | 0.002 |
| | optimization frequency | | 10 | |
| | evaluation frequency | | 10 | |
| | evaluation batch size | | 32 | |
| | evaluation step | | 1 | |
| | prediction reward weight | | 1.0 | |
| ECL−Sco | coefficient | 0.002 | 0.02 | 0.001 |
| | regularization starts after N steps | 100K | 100K | 750K |

four distinct categories of latent state variables by IFactor to conduct empowerment maximization for policy learning. Moreover, we follow the same parameter settings in IFactor, and used the same video background in all tasks.

Table 5: Hyperparameters for downstream task learning in all environments.

| Method | hyperparameters | Environments | | |
|---|---|---|---|---|
| | | Chemical | Physical | Manipulation |
| Reward Predictor | training steps | 300K | 1.5M | 2M |
| | optimizer | | Adam | |
| | learing rate | | 3e-4 | |
| | batch size | | 32 | |
| CEM | number of candidates | 64 | | 128 |
| | number of iterations | 5 | | 10 |
| | number of top candidates | | 32 | |
| | action_noise | | 0.03 | |

## D.3 RESULTS OF CAUSAL DYNAMICS LEARNING

We compare the performance of causal dynamics learning with score-based method GRADER (Ding et al., 2022), CDL (Wang et al., 2022c) and constraint-based method REG (Wang et al., 2021) across different environments. The experimental results, presented in Table 6, reveal that although

Table 6: Compared results of causal graph learning on three chemical and physical environments.

| Metrics | Methods | Chain | Collider | Full | Physical |
|---------|---------|-------|----------|------|----------|
| **Accuracy** | `ECL-Con` | **1.00±0.00** | **1.00±0.00** | **1.00±0.00** | **1.00±0.00** |
| | `ECL-Sco` | 0.99±0.00 | 0.99±0.00 | 0.99±0.01 | **1.00±0.00** |
| | GRADER | 0.99±0.00 | 0.99±0.00 | 0.99±0.00 | - |
| **Recall** | `ECL-Con` | **1.00±0.00** | **1.00±0.00** | **0.97±0.00** | **1.00±0.00** |
| | `ECL-Sco` | **1.00±0.00** | 0.99±0.01 | 0.90±0.02 | **1.00±0.00** |
| | GRADER | 0.96±0.03 | 0.99±0.02 | 0.96±0.02 | - |
| **Precision** | `ECL-Con` | **1.00±0.00** | **1.00±0.00** | 0.96±0.02 | **1.00±0.00** |
| | `ECL-Sco` | 0.99±0.01 | 0.99±0.01 | 0.97±0.03 | **1.00±0.00** |
| | GRADER | 0.94±0.04 | 0.90±0.05 | **1.00±0.00** | - |
| **F1 Score** | `ECL-Con` | **1.00±0.00** | **1.00±0.00** | 0.97±0.01 | **1.00±0.00** |
| | `ECL-Sco` | 0.99±0.00 | 0.99±0.00 | 0.93±0.02 | **1.00±0.00** |
| | GRADER | 0.95±0.03 | 0.94±0.03 | **0.98±0.01** | - |
| **ROC AUC** | `ECL-Con` | **1.00±0.00** | **1.00±0.00** | **0.98±0.01** | **1.00±0.00** |
| | `ECL-Sco` | 0.99±0.01 | 0.99±0.01 | 0.95±0.01 | **1.00±0.00** |
| | GRADER | 0.94±0.02 | 0.99±0.01 | 0.96±0.01 | - |

GRADER exhibits superior performance in the chemical full environment, `ECL`-based methods overall achieve better results than GRADER across three chemical environments. In the accuracy assessment metrics, `ECL-Con` attains 100% precision, and across the chain and collider environments, all evaluation metrics achieve perfect 100% scores. Furthermore, in the physical environment, our proposed methods attain 100% performance. The result of rigorous evaluation metrics substantiates that incorporating `ECL` has boosted the dynamics model performance. These experimental results further validate the effectiveness of the proposed `ECL` approach in both sparse and dense modal environments.

Furthermore, we analyze the prediction accuracy performance of the causal dynamics constructed by our proposed method. The multi-step (1-5 steps) prediction experimental results across four environments are illustrated in Figure 11. `ECL-Con` and CDL exhibit smaller declines in accuracy as the prediction steps increase, benefiting from the causal discovery realized based on conditional mutual information. Compared to REG, `ECL-Sco` achieves a significant improvement in accuracy under different settings. Concurrently, we find that the outstanding out-of-distribution experimental results further corroborate the strong generalization capability of our proposed method. By actively leveraging the learned causal structure for empowerment-driven exploration, `ECL` facilitates more accurate causal discovery. Overall, we can demonstrate that the proposed `ECL` framework realizes efficient and robust causal dynamics learning.

## D.4 VISUALIZATION ON THE LEARNED CAUSAL GRAPHS

We conduct a detailed comparative analysis by visualizing the learned causal graphs. In each causal graph, these are $d_{\mathcal{S}}$ rows and $d_{\mathcal{S}} + 1$ columns, and the element at the $j$-th row and $i$-th column represents whether the variable $s_{t+1}^j$ depends on the variable $s_{t+1}^i$ if $j < d_{\mathcal{S}} + 1$ or $a_t$ if $j = d_{\mathcal{S}} + 1$, measured by CMI for score-based methods and Bernoulli success probability for Reg. First, the causal graph learning scenario in the chemical chain environment is shown in Figure 12. Compared to CDL and REG, `ECL-Con` accurately uncovers the causal relationships among crucial elements, such as all different dimensions between states and actions, outperforming the other two methods. Moreover, we achieve extensive elimination of causality between irrelevant factors. These results demonstrate the accuracy of the proposed method in causal inference within the chemical chain environment.

Furthermore, for the chemical collider environment, the compared causal graphs are depicted in Figure 13. We can observe that both CDL and `ECL-Con` achieved optimal discovery of causal relationships. Moreover, in contrast to the REG method, `ECL-Con` is not impeded by interference from irrelevant causal factors. For the chemical full environment, the causal graph is illustrated in

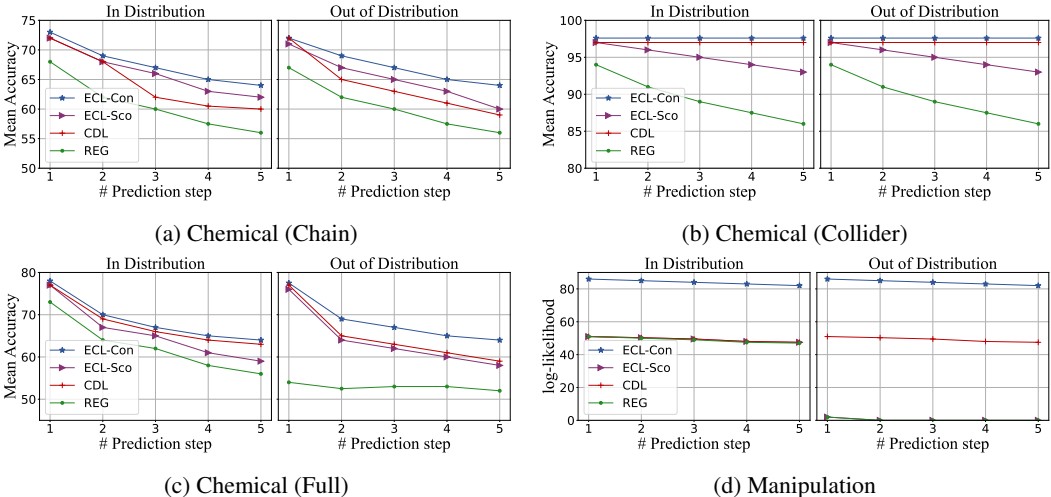

Figure 11: Multi-step prediction performance for four basic environments. (**Left**) prediction on in distribution states. (**Right**) prediction on OOD states.

Figure 14. Compared to CDL, `ECL-Con` better excludes interference from irrelevant causal factors. In comparison with the REG method, `ECL-Con` attains superior overall performance in discovering causal relationships. Additionally, `ECL-Con` reaches optimal learning performance when provided the true causal graph.

Moreover, for the manipulation environment, the experimental results are presented in Figures 15 and 16. From the results in Figure 6, we can discern that `ECL-Con` achieves around 90% overall fitting degree with the true causal graph and accurately learns the causal association between state and action. Compared to CDL shown in Figure 16, `ECL-Con` learns more causal associations from relevant causal components related to the gripper, movable states, and actions. Conversely, in contrast to REG, `ECL-Con` better excludes interference from irrelevant causal factors, such as unmovable and marker states. In summary, the proposed method achieves more accurate and efficient learning performance in causal dynamics learning. In the subsequent section, we will delve further into analyzing the enhanced performance of `ECL` in optimizing causal dynamics and reward models, and how these optimizations manifest in the learning policies for downstream tasks, including complex pixel-based tasks.

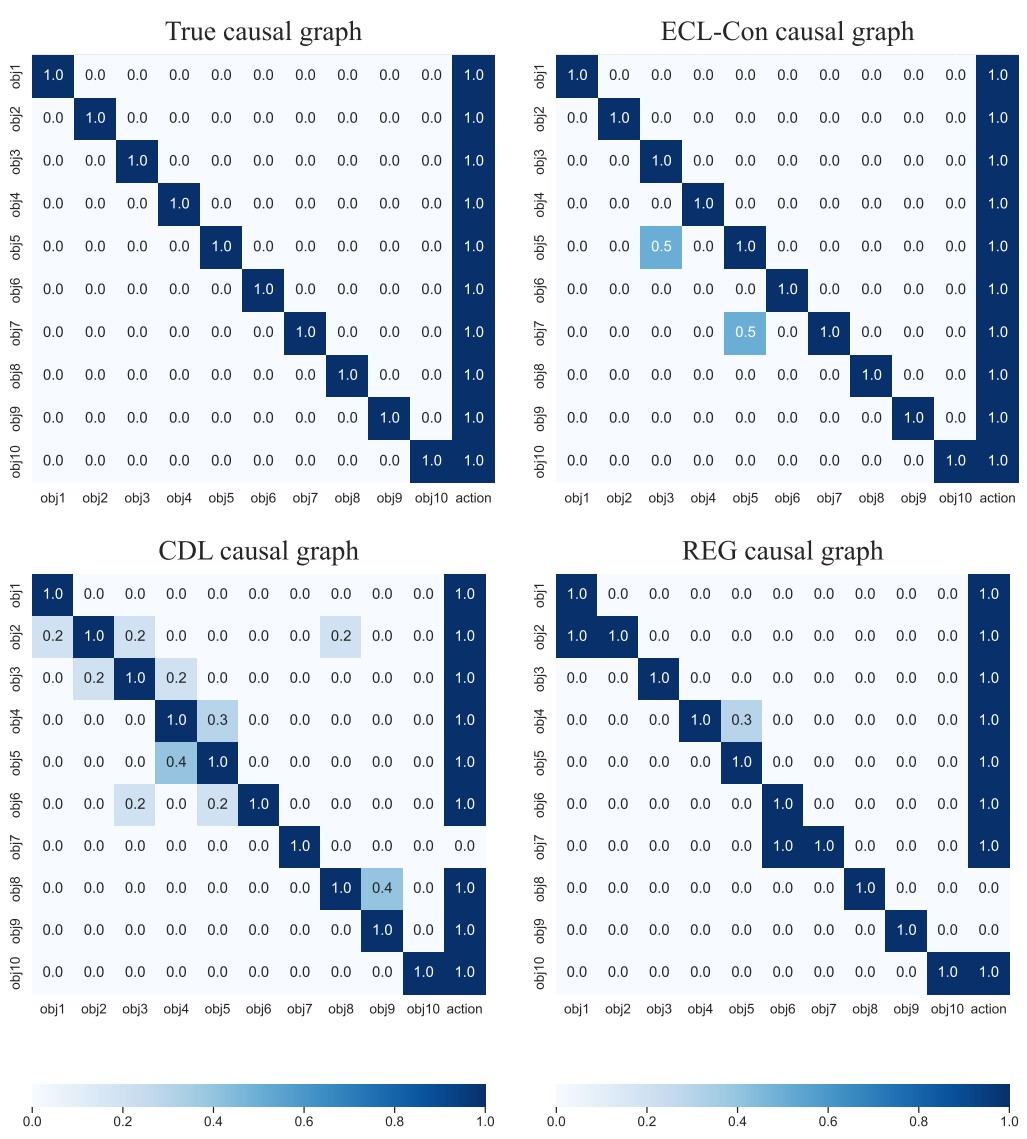

Figure 12: Causal graph for the chemical chain environment learned by the `ECL`, CDL and REG.

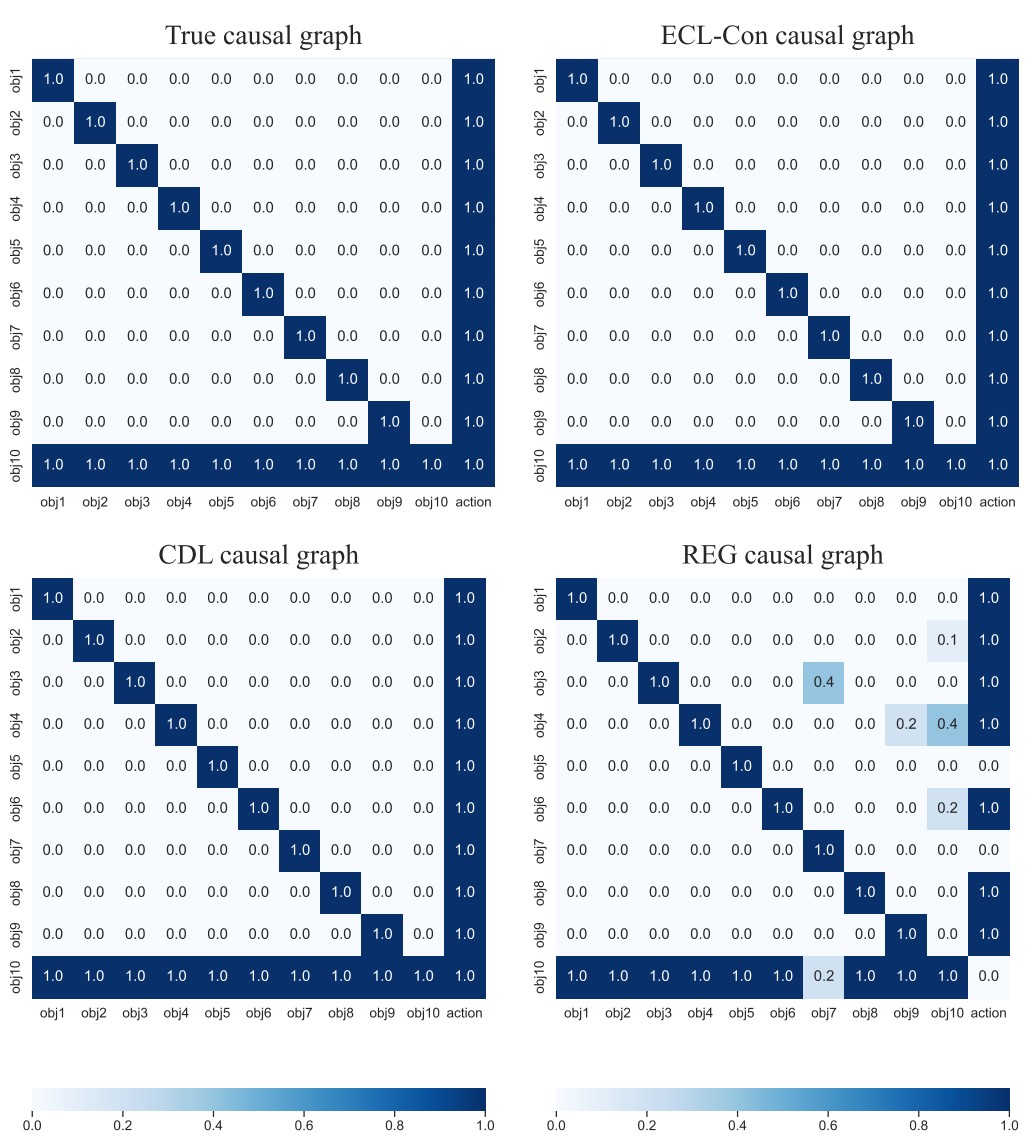

Figure 13: Causal graph for the chemical collider environment learned by the **ECL**, CDL and REG.

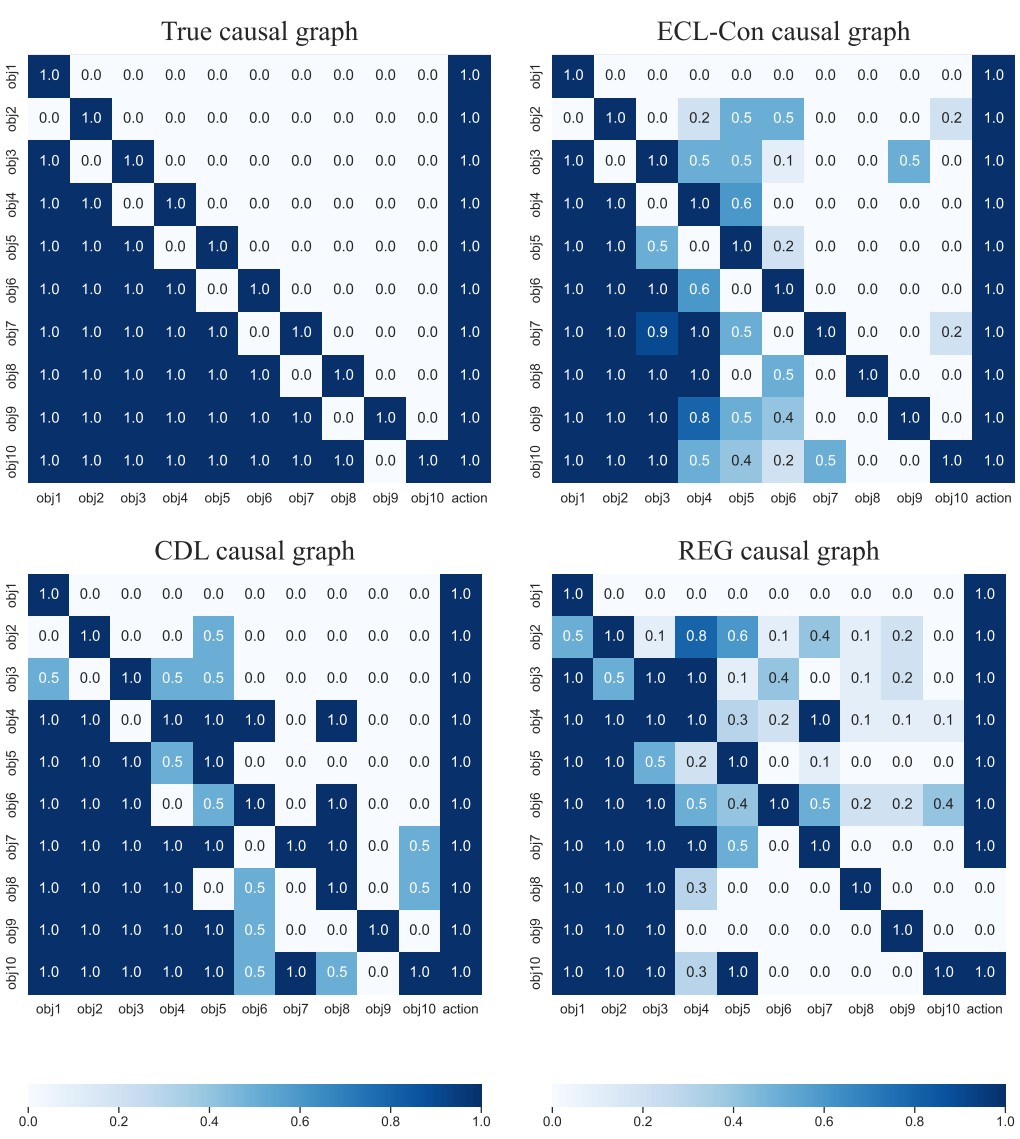

Figure 14: Causal graph for the chemical full environment learned by the **ECL**, CDL and REG.

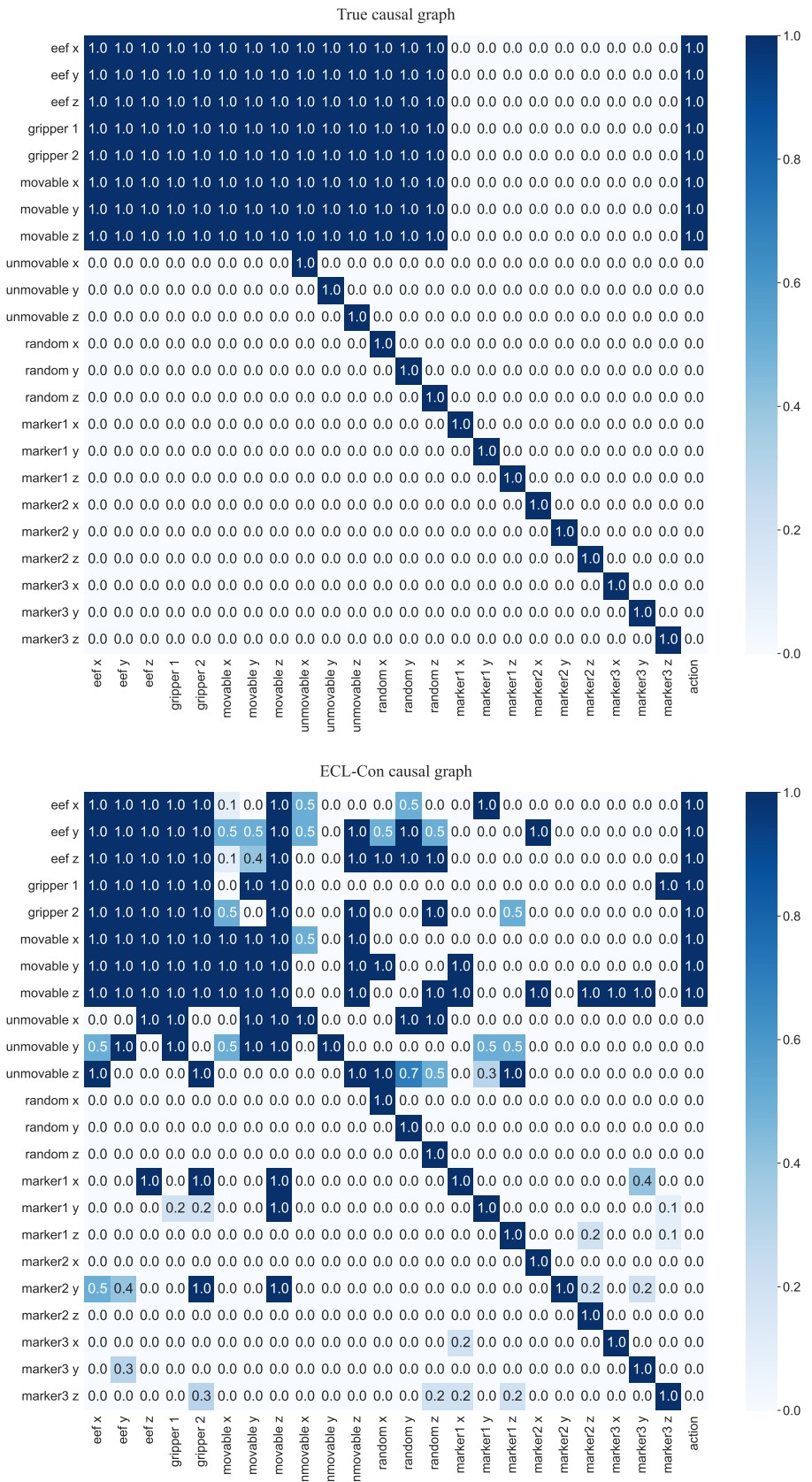

Figure 15: Causal graph for the manipulation environment learned by the true graph and **ECL**.

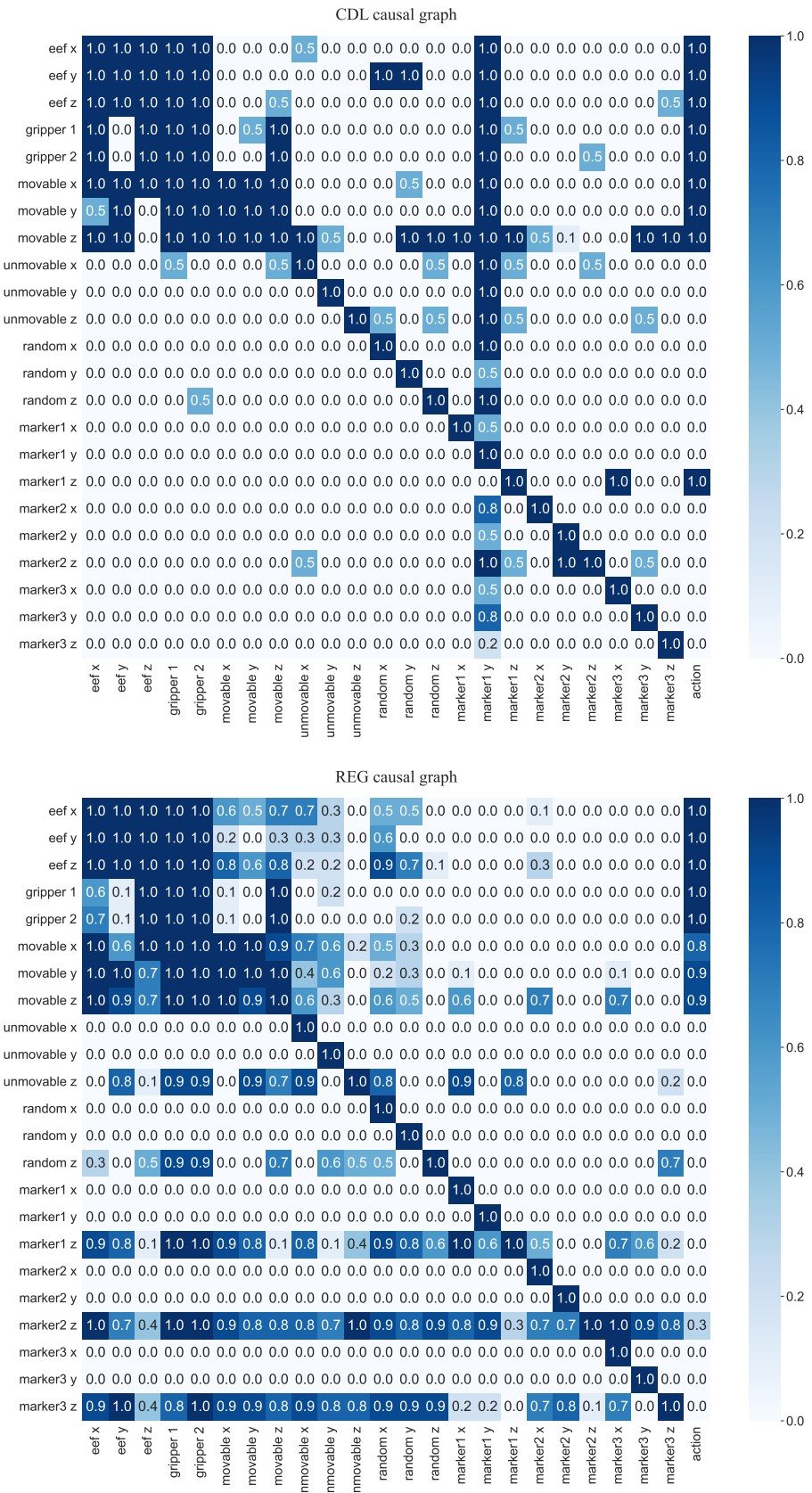

Figure 16: Causal graph for the manipulation environment learned by CDL and REG.

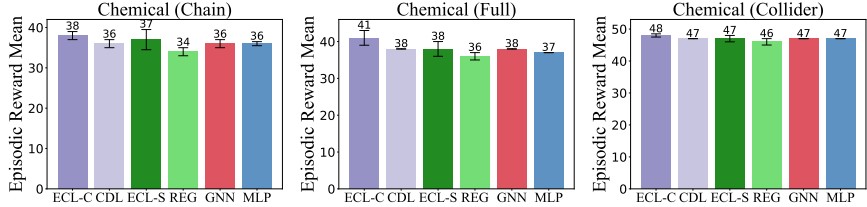

Figure 17: The task learning of episodic reward in three environments with `ECL-Con` (ECL-C), `ECL-Sco` (ECL-S) and baselines.

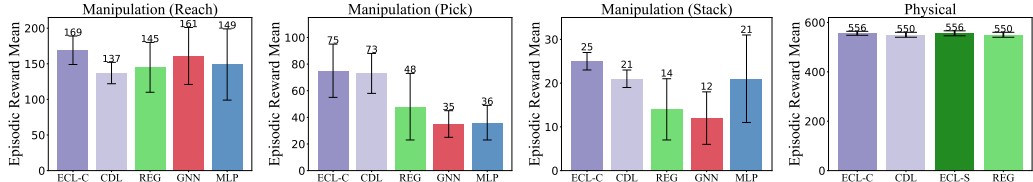

Figure 18: The task learning of episodic reward in three manipulation and physical environments.

## D.5 DOWNSTREAM TASKS LEARNING

As illustrated in Figures 17 and 18, `ECL-Con` attains the highest reward across three environments when compared to dense models like GNN and MLP, as well as causal approaches such as CDL and REG. Notably, `ECL-Con` outperforms other methods in intricate manipulation tasks. Furthermore, `ECL-Sco` surpasses REG, enhancing model performance and achieving a reward comparable to CDL. The proposed curiosity reward encourages exploration and avoids local optimality during the policy learning process. Moreover, `ECL` excels not only in accurately uncovering causal relationships but also in enabling efficient learning for downstream tasks.

**Sample efficiency analysis.** We perform comparative analysis of downstream tasks learning across all environments. As depicted in Figure 19 for experiments in three chemical environments, we can find that `ECL-Con` and `ECL-Sco` achieve outstanding performance in all three environments. Furthermore, the policy learning exhibits relative stability, reaching a steady state after approximately 400 episodes. Additionally, Figure 20 illustrates the reward learning scenarios in the other four environments. Within the intricate manipulation environment, `ECL-Con` facilitates more expeditious policy learning. Moreover, in the dense physical environment, `ECL-Con` and `ECL-Sco` also exhibit the most expeditious learning efficiency. The experimental results demonstrate that the proposed methods outperform CDL. Moreover, compared to CDL, `ECL` enhances sample efficiency, further corroborating the effectiveness of the proposed intrinsic-motivated empowerment method.

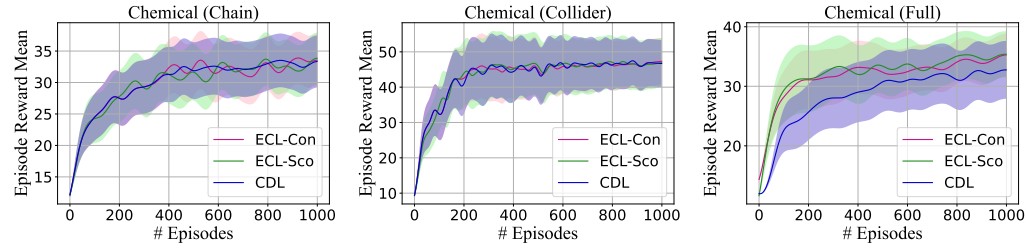

Figure 19: The task learning curves of episodic reward in three chemical environments and the shadow is the standard error.

**Causal Discovery with FCIT** We further conduct causal discovery using the explicit conditional independence test, specifically the Fast Conditional Independence Test (FCIT) employed in GRADER (Ding et al., 2022), for task learning evaluation. The comparative task learning results

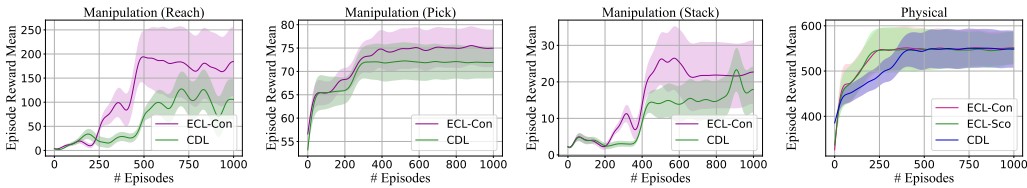

Figure 20: The task learning curves of episodic reward in four environments and the shadow is the standard error.

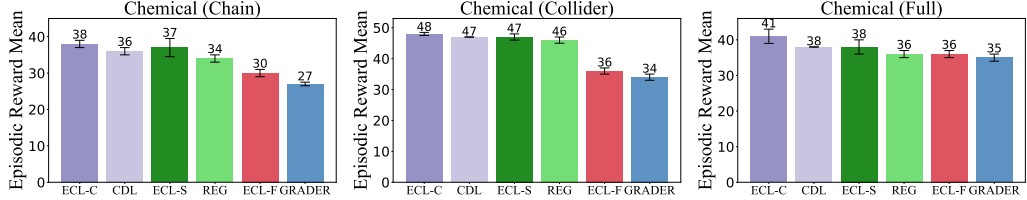

Figure 21: The task learning of episodic reward in three chemical environments. **ECL-S** represents **ECL** with score-based causal discovery. **ECL-C** represents **ECL** with L1-norm regularization of constraint-based causal discovery. **ECL-F** represents **ECL** with FCIT (used in GRADER for causal discovery).

are presented in Figure 21. These findings demonstrate that **ECL-FCIT**, achieves improved policy learning performance than GRADER, further validating the effectiveness of our proposed learning framework **ECL**.

### D.6 PIXEL-BASED TASKS LEARNING

We evaluate **ECL** on 5 pixel-input tasks across 3 latent state environments. Figure 22 presents comparative experimental results and visualized trajectories in the modified cartpole task. Our findings reveal that **ECL** achieves superior sample efficiency compared to IFactor. Furthermore, the visualized results demonstrate **ECL**'s effectiveness in controlling the target cartpole, successfully overcoming distractions from both the upper cartpole and the lower green light, which are not controlled in the IFactor policy.

Moreover, we conduct evaluations on three DMC tasks. The visualized results in Figure 23 confirm effective control for all three agents. Moreover, as shown in Figure 24, **ECL** achieves more stable average return results, corroborating the enhanced controllability provided by our proposed causal empowerment approach. Finally, we evaluate our method against DreamerV3 (Hafner et al., 2023), a current state-of-the-art approach, across three DMC tasks under noiseless settings. As shown in Figure 25, **ECL** consistently outperforms DreamerV3 in all 3 tasks.

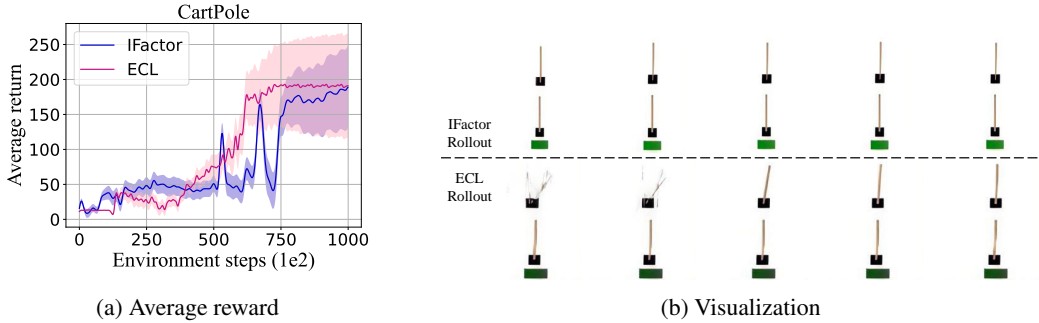

(a) Average reward           (b) Visualization

Figure 22: The results of average return compared with IFactor and visualized trajectories in Modified Cartpole environment.

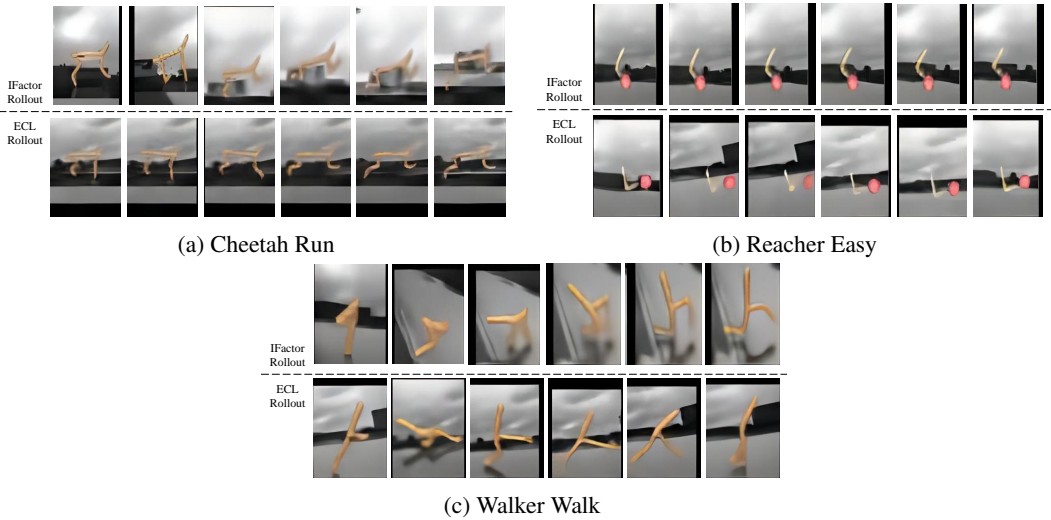

(a) Cheetah Run  (b) Reacher Easy

(c) Walker Walk

Figure 23: The results of visualization in three pixel-based tasks of DMC environment.

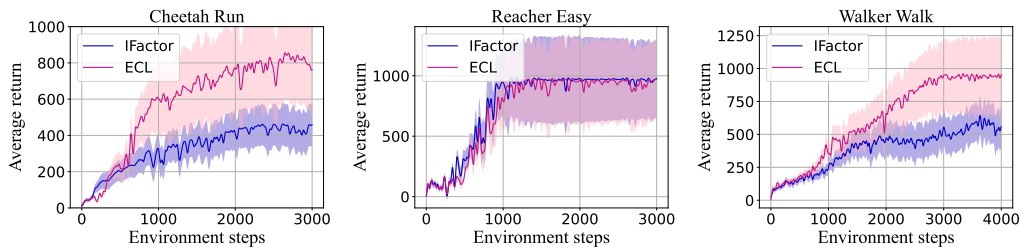

Figure 24: The results of average return compared with IFactor in three pixel-based tasks of DMC environment under video background setting.

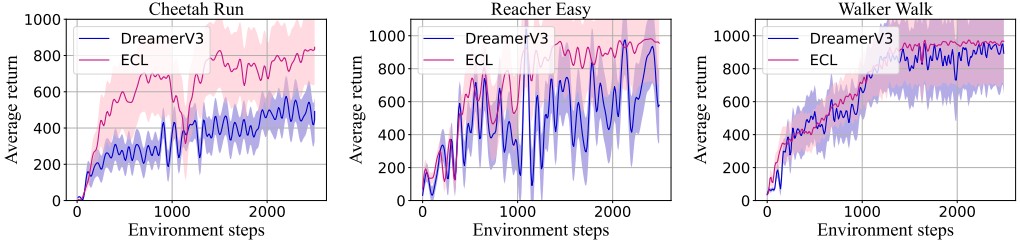

Figure 25: The results of average return compared with Dreamer in three pixel-based tasks of DMC environment under noiseless setting.

### D.7 PROPERTY ANALYSIS

**Training steps analysis.** For property analysis, we set different training steps for causal dynamics learning of **ECL-Con**. As depicted in Figure 26, in the chemical chain environment, we observe that the mean prediction accuracy reaches its peak at 300k training steps. A similar trend is observed in the collider environment, where the maximum accuracy is achieved at 150k training steps. Although in the full environment, **ECL** attains its maximum accuracy at 600k steps, which is higher than the 500k steps used for training CDL, we notice that at 500k steps, **ECL** has already achieved performance comparable to CDL. These results substantiate that our proposed causal action empowerment method effectively enhances sample efficiency and dynamics performance.

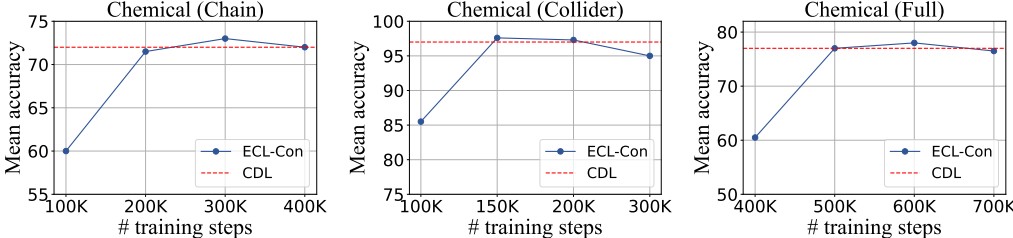

Figure 26: The mean accuracy of prediction with different training steps in chemical environments.

**Hyperparameter analysis.** We further analyze the impact of the hyperparameter $\lambda$ introduced in the downstream task reward function with CUR. We compare four different threshold settings, and the experimental results are depicted in Figure 27. From the results, we observe that when the parameter is set to 1, the policy learning performance is optimal. When the parameter is set to 0, the introduced curiosity cannot encourage exploratory behavior in the policy. Nonetheless, it still achieves reward performance comparable to CDL. This finding further corroborates the effectiveness of our method for dynamics learning. Conversely, when this parameter is set excessively high, it causes the policy to explore too broadly, subjecting it to increased risks, and thus more easily leading to policy divergence. Through comparative analysis, we ultimately set this parameter to 1. In our future work, we will further optimize the improvement scheme for the reward function.

**Computation cost.** To consider the computation cost, we calculate the computation time for two chemical tasks of Chain, and Collider. The experimental results shown in Figure 28 demonstrate that ECL achieves its performance improvements with minimal additional computational burden - specifically less than $10\%$ increase compared to CDL and REG. These results demonstrate that ECL's enhanced performance comes without significant computational cost. All experiments were conducted on the same computing platform with the same computational resources detailed in Appendix F.

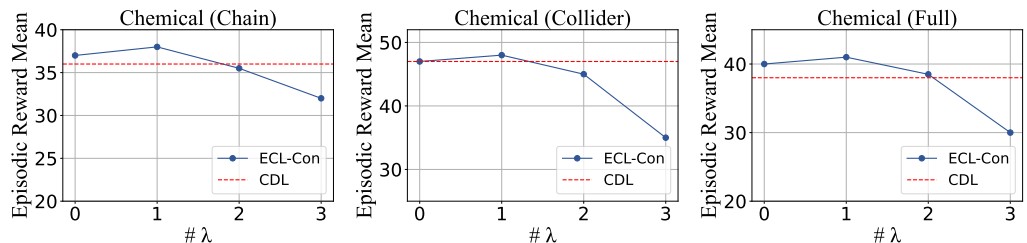

Figure 27: The episodic reward with different hyperparameter $\lambda$ in three chemical environments.

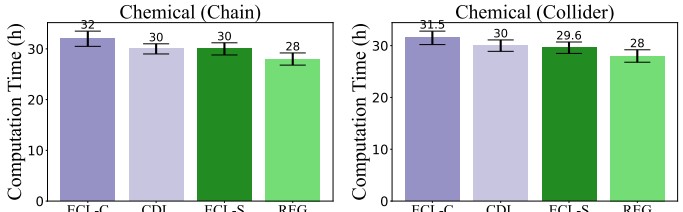

Figure 28: The computation time in two chemical environments.

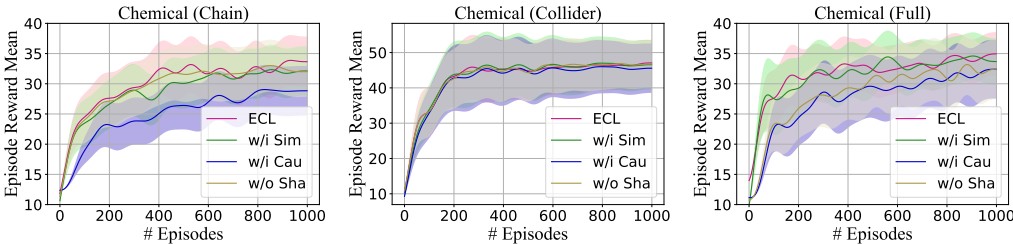

Figure 29: Learning curves of ablation studies in three chemical environments and the shadow is the standard error. w/ represents with. w/o represents without.

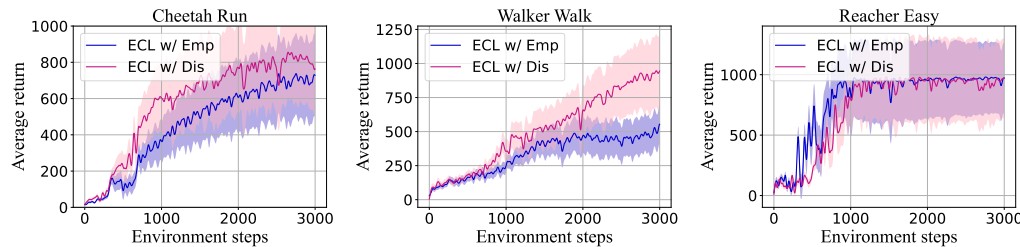

Figure 30: Learning curves of ablation studies in three DMC tasks and the shadow is the standard error. w/ represents with.

### D.8   ABLATION STUDIES

To further validate the effectiveness of the various components comprising the proposed **ECL** method, we designed a series of ablation experiments for verification. First, we implement the method without the first-stage model learning, simultaneously conducting causal model and task learning (w/ Sim) to verify the effectiveness of the proposed three-stage optimization framework. Second, we replace the curiosity reward introduced in the task learning with a causality motivation-driven reward (w/ Cau): $r_{\text{cau}} = \mathbb{E}_{(s_t, a_t, s_{t+1} \sim \mathcal{D})} \left[ \mathbb{KL} \left( P_{\text{env}} || P_{\phi_c, M} \right) - \mathbb{KL} \left( P_{\text{env}} || P_{\phi_c} \right) \right]$, and a method without reward shaping (w/o Sha), respectively, to verify the effectiveness of incorporating the curiosity reward.

The results presented in Figure 29 clearly demonstrate the superior performance of the **ECL** over all other comparative approaches. **ECL** achieves the highest reward scores among the evaluated methods. Moreover, when compared to the method with Sim, **ECL** not only attains higher cumulative rewards but also exhibits greater stability in its performance during training. Additionally, **ECL** significantly outperforms the methods with Cau and method without Sha, further highlighting the efficacy of our proposed curiosity-driven exploration strategy in mitigating overfitting issues. By encouraging the agent to explore novel states and gather diverse experiences, the curiosity mechanism effectively prevents the policy from becoming overly constrained.

We explore the difference between simply maximizing empowerment under the causal dynamics model (Eq. 8) versus maximizing the difference between causal and dense model empowerment (Eq. 10). Comparing **ECL** with empowerment (w/ Emp) against **ECL** with distance (w/ Dis) across three DMC tasks, our results in Figure 30 show that **ECL** w/ Dis achieves superior performance, and **ECL** w/ Emp also demonstrates strong learning capabilities.

Furthermore, we conducted comparative experiments between **ECL** and **ECL** without curiosity reward. The learning curves for episodic reward and success rate, shown in Figure 31, demonstrate that the curiosity reward plays a crucial role in preventing policy overfitting during the learning process. We also carried out experiments with different values of $\lambda$. The success rate shown in Figure 31 shows the effectiveness of the curiosity reward.

In summary, **ECL** facilitates effective and controllable policy learning for agents operating in complex environments. The curiosity-driven reward enables the agent to acquire a comprehensive understand-

ing of the environment while simultaneously optimizing for the desired task objectives, resulting in superior performance and improved sample efficiency.

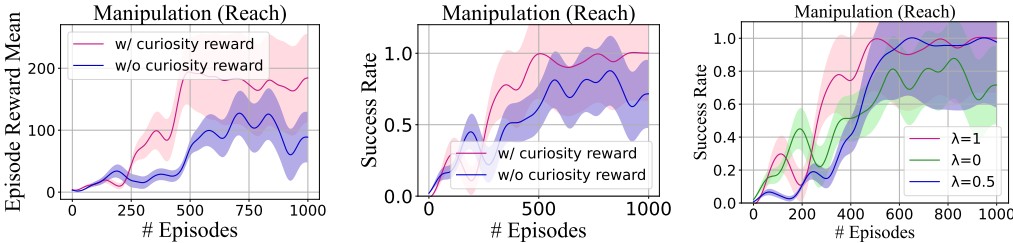

Figure 31: Learning curves of ECL with and without curiosity reward in manipulation reach task, and with different $\lambda$ settings. The shadow is the standard error. w/ represents with.

# E    DETAILS ON THE PROPOSED FRAMEWORK

Algorithm 1 lists the full pipeline of **ECL** below.

# F    EXPERIMENTAL PLATFORMS AND LICENSES

## F.1    PLATFORMS

All experiments of this approach are implemented on 2 Intel(R) Xeon(R) Gold 6444Y and 4 NVIDIA RTX A6000 GPUs.

## F.2    LICENSES

In our code, we have utilized the following libraries, each covered by its respective license agreements:

- PyTorch (BSD 3-Clause "New" or "Revised" License)
- Numpy (BSD 3-Clause "New" or "Revised" License)
- Tensorflow (Apache License 2.0)
- Robosuite (MIT License)
- CausalMBRL (MIT License)
- OpenAI Gym (MIT License)
- RoboDesk (Apache License 2.0)
- Deep Mind Control (Apache License 2.0)

---

**Algorithm 1** Empowerment through causal structure learning for model-based RL

---

**Input**: policy network $\pi_e$, $\pi_\theta$, transition collect policy $\pi_{\text{collect}}$, epoch length of dynamics model training, causal empowerment and downstream task policy learning $H_{\text{dyn}}$, $H_{\text{emp}}$, and $H_{\text{task}}$, evaluation frequency for causal mask learning $f_{\text{eval}}$

---

**Step 1: Model Learning**

**for** each environment step $t$ **do**
    Collect transitions $\{(s_i, a_i, r_i, s_i')\}_{i=1}^{|\mathcal{D}_{\text{env}}|}$ with $\pi_{\text{collect}}$ from environment
    Add transitions to replay buffer $\mathcal{D}_{\text{collect}}$
**end for**
**for** $epoch = 1, \cdots, H_{\text{dyn}}$ **do**
    Sample transitions $\{(s_i, a_i, s_i')\}_{i=1}^{|\mathcal{D}_{\text{dyn}}|}$ from $\mathcal{D}_{\text{collect}}$
    Train dynamics model $P_{\phi_c}$ with $\{(s_i, a_i, s_i')\}_{i=1}^{|\mathcal{D}_{\text{dyn}}|}$ followed Eq. 4
    **if** $epoch \% f_{\text{eval}} == 0$ **then**
        Sample transitions $\{(s_i, a_i, s_i')\}_{i=1}^{|\mathcal{D}_{\text{cau}}|}$ from $\mathcal{D}_{\text{collect}}$
        Learn causal dynamics model with causal mask using different causal discovery
        methods followed Eq. 5
    **end if**
    Sample transitions $\{(s_i, a_i, r_i, s_i')\}_{i=1}^{|\mathcal{D}_{\text{rew}}|}$ from $\mathcal{D}_{\text{collect}}$
    Train reward model $P_{\varphi_r}$ with $\{(s_i, a_i, r_i, s_i')\}_{i=1}^{|\mathcal{D}_{\text{rew}}|}$ and $\phi_c(\cdot \mid M)$ followed Eq. 6
**end for**

**Step 2: Model Optimization**

Collect transitions $\{(s_i, a_i, r_i, s_i')\}_{i=1}^{|\mathcal{D}_{\text{emp}}|}$ with policy $\pi_e$
**for** $epoch = 1, \cdots, H_{\text{emp}}$ **do**
    Maximize $(\mathcal{E}_{\phi_c}(s_{t+1} \mid M) - \mathcal{E}_{\phi_c}(s_{t+1}))$ with transitions sampled from $\mathcal{D}_{\text{emp}}$ for policy $\pi_e$ learning
    Add transitions sampled with $\pi_e$ to $\mathcal{D}_{\text{emp}}$
    **if** $epoch \% f_{\text{eval}} == 0$ **then**
        Optimize causal mask $M$ and reward model with transitions sampled from $\mathcal{D}_{\text{emp}}$
        followed Eq. 5 and Eq. 6
    **end if**
**end for**

**Step 3: Policy Learning**

**for** $epoch = 1, \cdots, H_{\text{task}}$ **do**
    Collect transitions $\{(s_i, a_i, r_i, s_i')\}_{i=1}^{|\mathcal{D}_{\text{task}}|}$ with $\pi_\theta$
    Compute predicted rewards $r_{\text{task}}$ by learned reward predictor
    Calculate curiosity reward $r_{\text{cur}}$ by Eq. 11
    Calculate $r \leftarrow r_{\text{task}} + \lambda r_{\text{cur}}$
    Optimize policy $\pi_\theta$ by the CEM planning
**end for**
**return** policy $\pi_\theta$

