# OpenReview forum: "Towards Empowerment Gain through Causal Structure Learning in Model-Based Reinforcement Learning"
_ICLR.cc/2025/Conference — ICLR 2025 Poster_

### Official Review · Reviewer_4AH7 · 2024-11-01

**Soundness:** 3
**Presentation:** 2
**Contribution:** 3
**Rating:** 6
**Confidence:** 4

**Summary:**

The authors propose ECL (Empowerment through Causal Learning) that has two main components: (1) a causal dynamics model of the environment that is learned from data and (2) a mechanism to "empower the causal structure for exploration, simultaneously using data gathered through exploration to update the causal dynamics model". The objective of the causal structure for exploration is to obtain a dynamics model that "could be more controllable than dynamics models without the causal structure". On top of this, an intrinsic curiosity reward is developed "to mitigate overfitting during downstream task learning".


--
Given some improvements made to the paper, I increase my score to 5.
--
Given some additional improvements made to the paper, I increase my score to 6 (see comments below)

**Strengths:**

The objective of learning a model with some causal structure that can be used for instance in the context of exploration is an important research topic.

**Weaknesses:**

Main weaknesses:
- Notations are not clearly defined. For instance, the causal mask $M$ is introduced (i) without clear mathematical definition and (ii) it's not $M$ that is used later but $M^{s \rightarow s'}$ in Equation 5 (see lines 204 to 215). This makes the methodology unclear.
- The vocabulary does not relate to clearly defined concepts, e.g. "causal understanding", "causal reasoning" in lines 153-155.

Additional comments:
- Key elements are described in the appendix instead of the main text, e.g. end of page 4: the causal loss that represents "the objective term associated with learning the causal structure" are given in Appendix D.2.

**Questions:**

- How can a curiosity-based reward prevent "overfitting during task learning" (lines 270-272)?
- Figure 5: is it the result of only one seed? Why is there only one seed used?

---

> ### Author Response · Authors · 2024-11-20
> **Response to Reviewer 4AH7**
>
> We thank you for your constructive suggestions and detailed comments. We have provided the response and updated the revision.
>
> > **Q1**: Notations are not clearly defined. For instance, the causal mask M is introduced (i) without clear mathematical definition and (ii) it's not M that is used later but Ms\rightarows′  in Equation 5 (see lines 204 to 215). This makes the methodology unclear.
>
> **R1**: The causal mask M provides a general representation of causal structures. In ECL, we learn two specific causal masks - $M^{s \to s}$ and $M^{a \to s}$  based on the MDP framework to optimize both dynamics and policy learning. We have revised Section 3.1 (lines 201-203) to provide a clearer mathematical definition and description of the causal mask and its relationship in the revised Sections 2.1 and 3.1.
>
> Furthermore, we have provided a notation table in Appendix Table 2, where the essential variables and parameters in the objective functions are given (together with details and remarks). We also illustrate the trainable parameters in each objective function (marked in red in revised Appendix C.2).
>
> **Related Revised Section: Section 2.1 (lines 138-142) and 3.1 (lines 201-203), Appendix Table 2, Appendix C.2.**
>
> > **Q2**: The vocabulary does not relate to clearly defined concepts, e.g. "causal understanding", "causal reasoning" in lines 153-155.
>
> **R2**: We have revised the ambiguous terms, replacing "causal understanding" and "causal reasoning" with the more precise statement: "We aim to enhance the empowerment gain under the causal structure of the environment for improving controllability."
>
> **Related Revised Section: Section 2.2 (lines 155-156)**
>
> > **Q3**: Key elements are described in the appendix instead of the main text, e.g. end of page 4: the causal loss that represents "the objective term associated with learning the causal structure" are given in Appendix D.2.
>
> **R3**: We have moved the important content about the causal loss objective from Appendix D.2 into the main text of Section 3.1 for better accessibility.
>
> **Related Revised Section: Section 3.1 (lines 211-213)**
>
> > **Q4**: How can a curiosity-based reward prevent "overfitting during task learning" (lines 270-272)?
>
> **R4**: Overfitting may occur during steps 1 and 2 of causal discovery. To mitigate this, the curiosity-based reward is designed to encourage the agent to explore states that are challenging for the causal dynamics model to predict but are captured by the dense dynamics of the true environment. This incentivizes broader exploration, preventing the policy from becoming overly conservative and stuck in suboptimal behaviors.
>
> > **Q5**: Figure 5: is it the result of only one seed? Why is there only one seed used?
>
> **R5**: For statistical robustness, each experiment is conducted using **4 random seeds**. We have clarified this result in Section 5.2.1 of the revision.
>
> **Related Revised Section: Figure 5, Section 5.1**
>
> **We hope that these comments have addressed the reviewer’s concern. We are happy to answer any follow-up concerns. We thank the reviewer's comments, this helps us a lot to improve the paper!**

---

> > ### Author Response · Authors · 2024-11-25
> >
> > Thank you once again for your constructive comments and suggestions. We have provided responses and revisions to address your concerns, specifically regarding questions on notations and definitions (**R1**, **R2**, **R3**), clarification on the curiosity reward (**R4**, with additional evaluations on cases without this reward—see the general response and updated **Figure 31**), and clarifications on the random seeds used (**R5**). All corresponding changes have also been incorporated into the revised paper, with pointers provided in the response above.
> >
> > As the deadline approaches, please feel free to reach out if you have any further questions or concerns. We sincerely thank you once again for your time and effort!

---

> > > ### Comment · Reviewer_4AH7 · 2024-11-26
> > > **Thank you for the revision**
> > >
> > > I appreciate the feedback and changes made to the paper that has improved clarity.
> > >
> > > I'm willing to increase my score to 5. At the moment, I'm not willing to increase the score further because I believe that there are still some unclear elements. For instance,
> > > - In Equation 11, $r_{cur}$ seems to be defined in general instead of for a specific state-action pair. Can you clarify?
> > > - In the new version line 265, the reward model is denoted as $\phi_r$ but $P_{\varphi_r}$ is used earlier?
> > > - (minor) In Figure 4, the y axis is "episode reward". Should it be "episode return" or "sum of rewards over episode" ?

---

> > > > ### Author Response · Authors · 2024-11-26
> > > > **Thank you & further clarifications**
> > > >
> > > > Thank you for your response, as well as the additional pointers and suggestions. We are glad to hear that you believe the current version has improved in terms of clarity. Here we address the further points you raised, and all of these have been clarified or modified in the revised version.
> > > >
> > > > > In Equation 11, r_{cur} seems to be defined in general instead of for a specific state-action pair. Can you clarify?
> > > >
> > > > Yes, $r_{cur}$ is calculated over state-action pairs sampled from the collected data. To make it more clear, we have edited it as
> > > >
> > > >  $r_{\mathrm{cur}}=\mathbb{E}\_{(s_t, a_t, s_{t+1}) \sim \mathcal{D}}\left[{\mathbb{KL}}\left({P}\_{\rm{env}}(s_{t+1}|s_t, a_t)||{P}\_{{\phi_c},M}(s_{t+1}|s_t, a_t;\phi\_c, M)\right)-{\mathbb{KL}}\left({P}\_{\rm{env}}(s_{t+1}|s_t, a_t)||{P}\_{\phi_c}(s_{t+1}|s_t, a_t;\phi_c)\right)\right]$.
> > > >
> > > > We have also updated it in the revised main paper (Eq. 11).
> > > >
> > > > > In the new version line 265, the reward model is denoted as $\varphi_r$ but $P_{\varphi_r}$ is used earlier.
> > > >
> > > > Thanks for the pointer. We have clarified this notation in the revision (same line).
> > > >
> > > > > (minor) In Figure 4, the y-axis is "episode reward". Should it be "episode return" or "sum of rewards over episode" ?
> > > >
> > > > In these environments, we calculate the mean episode reward, following the metrics in baseline [1]. For better clarity, we changed the y aix from “episode reward” to “episode reward mean” in **all figures** (Figures 3, 4, 17, 18, 20, 21, 29, 31)  accordingly.
> > > >
> > > > [1] Wang, Zizhao, et al. "Causal Dynamics Learning for Task-Independent State Abstraction." International Conference on Machine Learning. PMLR, 2022.
> > > >
> > > >
> > > > ---
> > > > Thank you once again for your valuable suggestions, and the time and effort devoted to this review. All of these points have contributed to improving our manuscript. We have incorporated the changes in the current version, and we hope the clarity has been enhanced. Please feel free to let us know if you have any further questions. We would be happy to discuss and address them.

---

> > > > > ### Author Response · Authors · 2024-11-27
> > > > > **Thank you for the detailed review, and we are happy to discuss further concerns if any**
> > > > >
> > > > > Dear Reviewer 4AH7,
> > > > >
> > > > > Thank you for your detailed review and efforts. We have carefully addressed your concerns and made the necessary revisions. We have updated another revision that includes the modifications based on your feedback.
> > > > >
> > > > > As the revision deadline is approaching, please feel free to let us know if you have any additional concerns. We are happy to discuss and address them before the revision deadline. Alternatively, we are also happy to address any remaining points within this rebuttal thread before the discussion deadline.
> > > > >
> > > > > We sincerely appreciate your valuable suggestions and thank you once again for your thoughtful suggestions!
> > > > >
> > > > > Best,
> > > > >
> > > > > Submission 1429 Authors

---

> > > > > > ### Comment · Reviewer_4AH7 · 2024-11-27
> > > > > > **Thanks for the  changes**
> > > > > >
> > > > > > Thanks for the changes. I still do not understand why Equation 11 is not $r_{cur}(s,a)$ or $r_{cur}(s_t,a_t)$ instead of $r_{cur}$.
> > > > > >
> > > > > > There are quite a few details in the formalization and notations that do not seem fully right but that being said, I believe the paper provides valuable insights and my confidence is sufficient to increase my score to an accept recommendation (6).

---

> > > > > > > ### Author Response · Authors · 2024-11-27
> > > > > > > **Thank you!**
> > > > > > >
> > > > > > > Thank you for your further response and for recognizing our insights. To better clarify, we have updated the notation to  $r_{cur}(s,a)$. We sincerely appreciate your thoughtful and detailed review, particularly your comments on the clarity of the notations, which we have addressed in the revision.
> > > > > > >
> > > > > > > Once again, thank you for your constructive and encouraging feedback, as well as your time and effort in reviewing and discussion.

---

### Official Review · Reviewer_DR5q · 2024-11-01

**Soundness:** 3
**Presentation:** 4
**Contribution:** 3
**Rating:** 6
**Confidence:** 3

**Summary:**

This paper presents a novel framework named Empowerment through Causal Learning (ECL), which integrates causal structure learning with empowerment-driven exploration in Model-Based Reinforcement Learning (MBRL) by (1) causal model learning, (2) empowerment maximization, and dynamic model updating and (3) policy learning. The proposed method is agnostic to causal discovery methods and outperforms existing causal MBRL methods across several environments.

**Strengths:**

1. The writing is clear and the paper is in general easy to follow.
2. Actively applying causal discovery to learn environment dynamics (updating through newly collected data) is a novel approach.
3. The experiments cover a diverse set of environments, including state-based and pixel-based tasks. Both analysis on the learned causal dynamics and on the average return demonstrate substantial improvements compared to other methods and provide strong evidence of ECL across various tasks.

**Weaknesses:**

1. The framework aims at learning a consistent causal structure, thus cannot deal with scenarios when the causal dynamics change (change of number of objects, etc.) that might correspond to different behavior components. Maybe consider discuss potential extensions or modifications to their framework that could handle changing causal dynamics.
2. Other issues please refer to questions.

**Questions:**

1. In the motivating example, why do we merely focus on improving controllablility (row 2 and 3) and do not care about whether the agent find the true target? (L79-l87) And further, how to detect the "controllable" trajectories? Could you provide more specific details on how controllable trajectories are identified and measured?
2. While maximizing mutual information I(s;a), would it be helpful to also take into account the causal structure, i.e. maximizing the state and action dimensions that are dependent in the dynamics graph? Potential benefits and challenges of incorporating the related s,a dimension into the mutual information objective could be made clearer.
3. The result in Fig. 24 in DMC environment CHeetah and Walker seem to have not converged yet. Could the author compare ECL and IFactor when they both converge? I agree that the learning curve of ECL is already going up faster and steadier than that of IFactor, and I think it could also be helpful to see the convergence point of the policies.

---

> ### Author Response · Authors · 2024-11-20
> **Response to Reviewer DR5q (1/2)**
>
> We thank the reviewer for the insightful and encouraging comments, please see the following for our response.
>
> > **Q1**: The framework aims at learning a consistent causal structure, thus cannot deal with scenarios when the causal dynamics change (change of number of objects, etc.) that might correspond to different behavior components. Maybe consider discussing potential extensions or modifications to their framework that could handle changing causal dynamics.
>
> **R1**: Thank you for raising the important consideration about changing causal dynamics. In ECL, we have explored this through out-of-distribution settings by changing object positions, as detailed in Section 5.2.2. For the cases with changing dynamic functions or changing numbers of objects, we can extend ECL to handle these by incorporating local causal discovery [1] for dynamics learning or models for handling heterogeneous or nonstationary factors in causal discovery [2]. These can be added to the current framework, we will explore them in the future extension. We have added discussions in the future work for these points.
>
> [1] Hwang, Inwoo, et al. "Quantized Local Independence Discovery for Fine-Grained Causal Dynamics Learning in Reinforcement Learning." ICML 2024.
>
> [2] Huang, Biwei, et al. "Causal discovery from heterogeneous/nonstationary data." Journal of Machine Learning Research 21.89 (2020): 1-53.
>
> **Related Revised Sections: Section 7 (lines 531-539)**
>
> > **Q2**: In the motivating example, why do we merely focus on improving controllability (row 2 and 3) and do not care about whether the agent finds the true target? (L79-l87) And further, how to detect the "controllable" trajectories? Could you provide more specific details on how controllable trajectories are identified and measured?
>
> **R2**: Thank you for your suggestion. We focus on the true target through the reward feedback (by policy learning) while simultaneously improving controllability.
>
> We use the causal empowerment term $I(s_{t+1};a_t|M)$ (Eq.(3)) to measure the controllability. Higher values of this mutual information can somehow indicate greater controllability. The policy $\pi_e$ (Eq. (7)) is trained to maximize the causal empowerment and collects controllable trajectories which are then used to optimize both causal dynamics and reward models. Hence, by optimizing this target, the agent can identify the controllable representation and increase controllability over environments. We have clarified this statement in the revised Section 3.2.
>
> **Related Revised Section: Section 3.2 (lines 262-265)**
>
> > **Q3**: While maximizing mutual information I(s;a), would it be helpful to also take into account the causal structure, i.e. maximizing the state and action dimensions that are dependent on the dynamics graph? Potential benefits and challenges of incorporating the related s,a dimension into the mutual information objective could be made clearer.
>
> **R3**: We sincerely appreciate your insightful suggestion. The causal empowerment proposed in ECL actually aligns with your thinking. We intentionally integrate the causal mask $M^{a\to s}$ into the mutual information where the state is processed through this mask, which enhances empowerment gain under the learned causal structure.
>
> Moreover, to validate this approach, we also conducted experiments that directly maximize empowerment under the causal dynamics model (ECL w/ Emp) by maximizing $I(s_{t+1};a_t|M)$ without maximizing the difference between causal and dense models as used in Eq. (7). Our results of average return below and learning curves in Figure 30 of Appendix D.8 in the revision show that ECL w/ Emp still achieves better learning performance than baseline IFactor while not quite matching the original objective Eq. (7) used in ECL.
>
> |             | CheetahRun | ReacherEasy | WalkerWalk |
> |-------------|------------|-------------|------------|
> | ECL | 797±87     | 981±5       | 963±25     |
> | ECL w/ Emp | 707±47     | 974±26      | 545±27   |
> | IFactor | 461±26     | 965±23     | 545±51     |
>
> **Related Revised Section: Figure 30, Appendix D.8 (line 1823-1827)**

---

> ### Author Response · Authors · 2024-11-20
> **Response to Reviewer DR5q (2/2)**
>
> > **Q4**: The result in Fig. 24 in DMC environment CHeetah and Walker seem to have not converged yet. Could the author compare ECL and IFactor when they both converge? I agree that the learning curve of ECL is already going up faster and steadier than that of IFactor, and I think it could also be helpful to see the convergence point of the policies.
>
> **R4**: Thank you for your valuable suggestion. Following your suggestion, we extended our experiments with more environmental steps, reflected in the revised Figure 24 of Appendix D.6 and the results of the average return shown below. The updated results show that even at convergence, ECL maintains its better performance and sample efficiency than IFactor.
> | Method    | CheetahRun | ReacherEasy | WalkerWalk |
> |-----------|------------|-------------|------------|
> | ECL | 797±87     | 981±5       | 963±25     |
> | IFactor | 461±26     | 965±23     | 545±51     |
>
> **Related Revised Section: Figure 24, Appendix D.6**
>
> **We hope that these comments have addressed the reviewer’s concern. We are happy to answer any follow-up concerns. We thank the reviewer's comments, this helps us a lot to improve the paper!**

---

> > ### Author Response · Authors · 2024-11-25
> >
> > Thank you once again for your insightful and encouraging comments! We have provided responses above, particularly related to the clarifications on the framework (**R1**, **R2**, **R3**) and a full evaluation of the comparison with IFactor (**R4**). Clarifications and new results have also been incorporated into the revised paper, with pointers provided in the response.
> >
> > If you have any further questions, please feel free to let us know. We would be more than happy to discuss them before the discussion phase closes. Thank you once again for your time and effort!

---

### Official Review · Reviewer_8gmf · 2024-11-03

**Soundness:** 3
**Presentation:** 4
**Contribution:** 4
**Rating:** 8
**Confidence:** 5

**Summary:**

This paper introduces the empowerment through Causal Learning (ECL) framework to enhance controllability and learning efficiency in model-based RL. The framework combines empowerment, a measure of an agent’s ability to control its environment through causal structure learning. ECL enables agents to understand causal relationships within the environment, improving decision-making and policy learning. ECL was evaluated with different causal discovery methods across three environments, showing improved sample efficiency, accurate causal inference, and higher episodic rewards than other causal MBRL approaches​.

**Strengths:**

- Overall, I think this is a good paper. The idea is interesting. The authors provide enough details and explanations in the technique section. The experiments are thorough and enough to demonstrate the advantages of ECL.
- ECL combines causal learning with empowerment-driven exploration, which is novel. By leveraging causal structures, the model enables agents to control their environments more effectively and make informed decisions, adding depth to RL's traditional empowerment approach. Through empowerment-driven exploration, ECL enhances the agent’s ability to efficiently sample relevant experiences, reducing the data requirements compared to conventional MBRL methods. This leads to faster learning and less dependence on extensive data.
- ECL has been tested across different environments, such as chemical, manipulation, and physical tasks, showing strong performance in sample efficiency, causal discovery accuracy, and episodic rewards.
- By incorporating a curiosity reward during policy learning, ECL encourages exploration while reducing the risk of overfitting specific causal structures. This helps the agent generalize better to new or out-of-distribution tasks.

**Weaknesses:**

Refer to the questions.

**Questions:**

1. Curiosity-driven exploration in RL is often sensitive and can be challenging to implement effectively. Are there different settings for setting curiosity in different experimental environments? Whether ECL is sensitive to curiosity rewards?
2. Given the complexity of the ECL structure, the ablation studies should not be omitted, i.e., ablations on reward design, basic model and other related parts should be added.

---

> ### Author Response · Authors · 2024-11-20
> **Response to Reviewer 8gmf**
>
> We thank you for the encouraging and insightful comments. All of them are invaluable for further improving our manuscript. Please refer to our response below.
>
> > **Q1**: Curiosity-driven exploration in RL is often sensitive and can be challenging to implement effectively. Are there different settings for setting curiosity in different experimental environments? Whether ECL is sensitive to curiosity rewards?
>
> **R1**: Thank you for your valuable feedback about curiosity-driven exploration. **We follow the same curiosity setting across different experimental environments.** For sensitivity analysis, we examined the impact of the hyperparameter $\lambda$ in the curiosity reward function across four threshold settings (0, 1, 2, 3). The experimental results of episodic reward across different environments are shown below. From these results, we observe that ECL maintains robust performance for $\lambda$ values under 3, with optimal policy learning achieved when $\lambda$ = 1.
> | $\lambda$ | chain | collider | full |
> |---|-------|----------|------|
> | 0 | 37    | 47       | 40   |
> | 1 | 38    | 48       | 41   |
> | 2 | 35.5  | 45       | 38.5 |
> | 3 | 32    | 35       | 30   |
>
> Furthermore, regarding curiosity design, we investigated a causality motivation-driven reward (ECL w/ Cau) that inverts the learning objective in step 3. Results in Figure 29 of Appendix D.8 in the revision demonstrate that ECL significantly outperforms this alternative approach. We also conducted a thorough examination of causal empowerment exploration, comparing directly maximizing the empowerment (Eq. (8)) given the causal dynamics model (ECL w/ Emp) versus empowerment distance maximization (Eq. (10)) used in the paper (ECL w/ Dis). Our findings in Figure 30 of Appendix D.8 in the revision reveal that while ECL w/ Emp demonstrates robust learning capabilities, ECL w/i Dis achieves superior performance.
>
> **Related Revised Section: Appendix D.7 (line 1752-1758) and Figure 29, Figure 30, D.8 (line 1823-1827)**
>
> > **Q2**: Given the complexity of the ECL structure, the ablation studies should not be omitted, i.e., ablations on reward design, basic model and other related parts should be added.
>
> **R2**: Thank you for your valuable suggestion. We have significantly expanded our ablation studies in Appendix D.8 of the revision. We examined different reward design variations for causal empowerment, including
> - (1). cases without maximizing the difference between causal and dense model (Eq. (10)) instead of maximizing causal empowerment (noted as w/ Emp in Appendix D.8) of Eq. (8) ,
> - (2). cases with the causality-driven reward $r\_{\mathrm{cau}}=\mathbb{E}\_{(s_t, a_t, s_{t+1} \sim \mathcal{D})}\left[{\mathbb{KL}}\left({P}\_{\rm{env}}||{P}\_{{\phi_c},M}\right)-{\mathbb{KL}}\left({P}\_{\rm{env}}||{P}\_{\phi_c}\right)\right], $ (noted as w/ Cau in Appendix D.8),
> - (3). cases without curiosity-driven reward $r_{\mathrm{cur}}$ (noted as w/ Sha in Appendix D.8);
> - (4). cases without the first-stage model learning, simultaneously conducting causal model and task learning (noted as w/ Sim in Appendix D.8) to verify the effectiveness of the proposed three-stage optimization framework.
>
> The detailed comparative results for these variations are presented in Figures 29 and 30 of Appendix D.8. We plan to further explore additional basic model combinations with ECL to enhance policy learning capabilities.
>
> **Related Revised Section: Figures 29, 30, Appendix D.8**

---

> > ### Author Response · Authors · 2024-11-25
> >
> > Thank you once again for your encouraging and constructive comments! We have provided responses to the points you raised, including ablations on hyperparameter choices, reward design, and related aspects. Please refer to the detailed list and results in the rebuttal, as well as the corresponding changes in the revised paper (pointers are provided in the response above).
> >
> > Additionally, regarding the role of the curiosity reward, we conducted new ablation studies in a complex manipulation environment. The results, including learning curves, are presented in Figure 31 of the appendix. These results demonstrate that for complex tasks, the curiosity reward plays a significant role in the framework.
> >
> > |                      | manipulation (Reach) |
> > |----------------------|----------------------|
> > | w/ curiosity reward  | 195.6±15.4           |
> > | w/o curiosity reward | 94.7±16.5            |
> >
> > Please do not hesitate to reach out if you have any further questions. Thank you once again for your time and effort!

---

> > > ### Comment · Reviewer_8gmf · 2024-11-26
> > > **Thank you for the response**
> > >
> > > I thank author for the response and additional experiments. I believe this is a good paper. I will keep my score.

---

> > > > ### Author Response · Authors · 2024-11-26
> > > > **Thank you!**
> > > >
> > > > Thank you for your encouraging response. We are glad that we were able to address your concerns. Your suggestions and comments have been invaluable in improving our work. Thank you again for your time and effort!

---

### Official Review · Reviewer_8bod · 2024-11-04

**Soundness:** 3
**Presentation:** 2
**Contribution:** 3
**Rating:** 6
**Confidence:** 3

**Summary:**

The authors present ECL, an agent that integrates empowerment with causal reasoning to better learn to control an environment, to then perform better in downstream tasks. First, ECL learns a causal dynamics model (dense dynamics model + causal mask) and reward model to maximize the likelihood of observed trajectories, with a regularization term on the causal mask encouraging the use of as few features as possible. Second, ECL alternates between updating a policy that leverages the causal mask to maximize empowerment, and using the data generated by running the policy to improve the causal mask and reward model. Finally, the learned models can be used to learn policies for downstream tasks, mitigating overfitting of the causal model with an intrinsic reward for observing transitions where the dense model fits better than the causal model. The authors demonstrate that ECL performs well in terms of both reward and sample efficiency compared to existing methods across environments, and accurately learns the true causal structure of the environments.

**Strengths:**

The approach appears to be novel. The integration of causal discovery, which by itself can be quite passive, with Empowerment, to emphasize controllability, is very interesting.

The authors tested on a wide spread of environments with different types of dynamics, showing impressive causal discovery and task performance.

**Weaknesses:**

The presentation of the algorithm is vague and unclear, with many technical details skimmed over. For instance:

 - The causally-factored MDP is difficult to understand and the explanations are very brief; it would be helpful to show the causal masks and adjacency matrices for the example environment in figure 1, and provide more explanation/intuition for what each term in the equations represents. See the questions section.

 - How the causal models are used for downstream tasks is not specified anywhere except a single line of the Appendix “the CEM planning”

The dynamics encoder predicts an independent probability for each feature of the next state given the current state and action- how realistic is this? On a similar vein, this method seems dependent on a well-defined feature space for the states and actions.

The choice of “standard MBRL” baselines to complement the causal baselines do not seem very standard- why not compare against state of the art MBRL algorithms such as [1]?

More minor writing issues:

 - The use of Dense dynamics model/dynamics encoder interchangeably to describe the same thing is confusing

 - Section 4 is quite repetitive of section 3

[1] Hafner, Danijar, et al. "Mastering diverse domains through world models." arXiv preprint arXiv:2301.04104 (2023).

**Questions:**

In the dynamics function f in equation (2), it’s not clear how exactly the adjacency matrices indicate the influence of current states and actions on the next state, what the output of each dot product represents and how they can be combined to get the ith dimension of the next state?

What is the relationship between the causal mask M and the adjacency matrices?

How can the mask be updated without also having to update the dynamics encoder? If the mask was very bad at Step 1, wouldn’t the dynamics encoder also be very suboptimal and not appropriate to continue using as the mask is improved in step 2?

Why does it not work to simply maximize the empowerment given the causal dynamics model, rather than the difference between that and the empowerment under the dense model?

It is not explained in section 3 where the reward from step 1 and 2 comes from. Section 4 describes a reward function formulated to select transitions covering more state-action pairs. How sensitive is the method to the design of this reward function? Reward functions can be notoriously hard to design; a lot of the difficulty of the problem might be obfuscated in this part.
In Step 3 of Algorithm 1 it says the learned reward predictor predicts $r_{task}$- how can it predict that if it was learnt during step 1 and 2 in the absence of any downstream tasks? And why are the $r_i$ in the transitions collected in line 2 ignored? Step 3 of section 4 implies that the causal model is only used to generate curiosity intrinsic rewards (which does not rely on the learned reward predictor at all) so this is inconsistent with Algorithm 1.

---

> ### Author Response · Authors · 2024-11-20
> **Response to Reviewer 8bod (1/3)**
>
> We thank the reviewer for the insightful and constructive feedback. We provide the response below.
>
> > **Q1**: How the causal models are used for downstream tasks is not specified anywhere except a single line of the Appendix “the CEM planning”
>
> **R1**: We appreciate your important question. The causal models are used to execute dynamic state transitions defined in Eq. (2). The causal models are utilized in conjunction with reward models for downstream tasks through CEM planning. Specifically, The causal model is a combined dynamic model with the causal structure that maximizes state transitions likelihood (Eq. (5)). The reward model evaluates these transitions and provides feedback in the form of rewards. The CEM handles the planning process by leveraging the predictions from the causal and reward models to optimize the task's objectives effectively. The causal masks have been used in all steps (See the visualized masks in Figure 2). We have clarified this process in Section 4 of the revision.
>
> **Related Revised Section: Section 4 (lines 302-306)**
>
> > **Q2**: The dynamics encoder predicts an independent probability for each feature of the next state given the current state and action- how realistic is this? In a similar vein, this method seems dependent on a well-defined feature space for the states and actions.
>
> **R2**: Thank you for the insightful question. It is not necessary to have well-defined features for our approach. In MDPs, where well-defined observable states are available, we can directly learn the graph based on those observations. For POMDPs, where raw states are not accessible and only high-dimensional, entangled observations (e.g., videos) are provided, we can still utilize encoders to extract low-dimensional states and learn the graph on these representations. Prior works, particularly in causal representation learning [1], have demonstrated that these low-dimensional states are identifiable, with a mapping from the estimated states to the true states existing under mild assumptions. While this aspect is not the focus of our work, we build on such findings (e.g., in our experiments on pixel-based benchmarks).
>
> Specifically, in our experiments: for chemical, manipulation, and physical environments, we utilize well-defined feature spaces for states and actions, which are explicitly designed for causal structure learning. For pixel-based environments such as DMC, Cartpole, and RoboDesk, ECL operates on latent states extracted by visual encoders. These encoders are supported by the identifiability theory proposed in IFactor [2], which ensures that these latent states can effectively map to the true states. While establishing identifiability is not the primary focus of our work, we leverage IFactor's encoders and include comparisons with IFactor in our experiments. Importantly, even in these settings, we can learn meaningful causal graphs. We have clarified this point in the revised Appendix D.2.
>
> [1] Schölkopf, Bernhard, et al. "Toward causal representation learning." Proceedings of the IEEE 109.5 (2021): 612-634.
>
> [2] Liu, Yuren, et al. "Learning world models with identifiable factorization." Advances in Neural Information Processing Systems 36 (2023): 31831-31864.
>
> **Related Revised Section: Appendix D.2 (lines 1110-1117)**
>
> > **Q3**: The choice of “standard MBRL” baselines to complement the causal baselines do not seem very standard- why not compare against state of the art MBRL algorithms such as [1]?
>
> **R3**: We're grateful for your suggestion about baseline comparisons. Following your feedback, we expanded our evaluation to include DreamerV3 [1], a state-of-the-art MBRL algorithm, testing it across three DMC tasks under noiseless settings. The results of average return are shown below and the detailed learning curves are shown in Figure 25 of the Appendix D.6 in the revision. These results demonstrate that ECL can achieve better performance and sample efficiency than DreamerV3.
> | Method    | CheetahRun | ReacherEasy | WalkerWalk |
> |-----------|------------|-------------|------------|
> | ECL       | 871±18     | 982±9       | 970±8      |
> | DreamerV3 | 519±46     | 584±138     | 862±80     |
>
> [1] Hafner, Danijar, et al. "Mastering diverse domains through world models." arXiv preprint arXiv:2301.04104 (2023).
>
> **Related Revised Section: Figure 25, Appendix D.6**
>
> > **Q4**: The use of Dense dynamics model/dynamics encoder interchangeably to describe the same thing is confusing.
>
> **R4**: Thank you for your valuable suggestion. We have unified these two statements in the revision.
>
>
> > **Q5**: Section 4 is quite repetitive of section 3
>
> **R5**: Thank you for your comment. The relationship between Sections 3 and 4 is intentional: Section 4 provides the practical implementation details of the theoretical formulation presented in Section 3. We have further simplified the practical implementation in Section 4.
>
> **Related Revised Section: Section 4**

---

> ### Author Response · Authors · 2024-11-20
> **Response to Reviewer 8bod (2/3)**
>
> > **Q6**: The causally-factored MDP is difficult to understand and the explanations are very brief; it would be helpful to show the causal masks and adjacency matrices for the example environment in figure 1, and provide more explanation/intuition for what each term in the equations represents. See the questions section.
>
> **R6**: Thank you for the suggestion. We have added a more detailed explanation of the causal factored structure using Figure 1 as an example (see the revised Section 1). For formal definitions, we recommend referring to Section 2.1. To ensure simplicity and intuition, Figure 1 represents a special case where the state dimension of each object is set to 1. Additionally, we have clarified the explanation of the equation in Section 2.
>
> **Related Revised Section: Section 1 (paragraph in red, on page 2), Section 2**
>
> > **Q7**: In the dynamics function f in equation (2), it’s not clear how exactly the adjacency matrices indicate the influence of current states and actions on the next state, what the output of each dot product represents and how they can be combined to get the ith dimension of the next state?
>
> **R7**: Regarding the dynamics function in equation (2), each dot product represents the causal effect between dimensional factors. For instance, in the adjacency matrices $M^{a\to s}$, each point (i,j) represents the causal effect from the i-th dimension of the action to the j-th dimension of the state. These causal effects are combined to determine each dimension of the next state.
>
> > **Q8**: What is the relationship between the causal mask M and the adjacency matrices?
>
> **R8**: They represent the same meaning in the causal structure. We have unified them in the revision.
>
> > **Q9**: How can the mask be updated without also having to update the dynamics encoder? If the mask was very bad at Step 1, wouldn’t the dynamics encoder also be very suboptimal and not appropriate to continue using as the mask is improved in step 2?
>
> **R9**: The causal mask M is optimized by maximizing $\mathcal{L}_{causal}$ (Eq.(5))​, while keeping the parameters of $\phi_c$  kept fixed during this learning step. If the mask is suboptimal in Step 1, the empowerment policy can implicitly refine the causal structure by iteratively maximizing the controllability of the environment while updating the causal mask (Eq. (7) and Step 2 in Figure 2). We have clarified this explanation in Section 3.2 of the revised manuscript.
>
> **Related Revised Section: Section 3.2 (lines 230-231)**
>
> > **Q10**: Why does it not work to simply maximize the empowerment given the causal dynamics model, rather than the difference between that and the empowerment under the dense model?
>
> **R10**: Thank you for raising this insightful point! We explore the difference between simply maximizing empowerment under the causal dynamics model (Eq. (8)) versus maximizing the difference between causal and dense model empowerment (Eq. (7)). By doing so, we aim to implicitly make the model with causal structures can provide better controllability than the dense model.
>
> To validate this, we also conducted experiments that directly maximize empowerment under the causal dynamics model (ECL w/ Emp) by maximizing $I(s_{t+1};a_t|M)$ without maximizing the difference between causal and dense models as used in Eq. (7). Our results of average return below and learning curves in Figure 30 of Appendix D.8 in the revision show that ECL w/ Emp still achieves better learning performance than baseline IFactor while not quite matching the original objective Eq. (7) used in ECL.
>
> |             | CheetahRun | ReacherEasy | WalkerWalk |
> |-------------|------------|-------------|------------|
> | ECL | 797±87     | 981±5       | 963±25     |
> | ECL w/ Emp | 707±47     | 974±26      | 545±27   |
> | IFactor | 461±26     | 965±23     | 545±51     |
>
> **Related Revised Section: Figure 30, Appendix D.8**
>
> > **Q11**: It is not explained in section 3 where the reward from step 1 and 2 comes from.
>
> **R11**: The reward is derived from the defined environmental feedback during online exploration when collecting trajectories.

---

> > ### Author Response · Authors · 2024-11-20
> > **Response to Reviewer 8bod (3/3)**
> >
> > > **Q12**: Section 4 describes a reward function formulated to select transitions covering more state-action pairs. How sensitive is the method to the design of this reward function? Reward functions can be notoriously hard to design; a lot of the difficulty of the problem might be obfuscated in this part.
> >
> > **R12**: We have performed the sensitivity analysis on the hyperparameter $\lambda$ in the curiosity reward function across four threshold settings (0,1,2,3). The experimental results of episodic reward across different environments are shown below. From these results, we observe that ECL maintains robust performance for $\lambda$ values under 3, with optimal policy learning achieved when $\lambda = 1$.
> >
> > | $\lambda $ | chain | collider | full |
> > |---|-------|----------|------|
> > | 0 | 37    | 47       | 40   |
> > | 1 | 38    | 48       | 41   |
> > | 2 | 35.5  | 45       | 38.5 |
> > | 3 | 32    | 35       | 30   |
> >
> > **Related Revised Section: Figure 27, Appendix D.7 (line 1752-1758)**
> >
> >
> > > **Q13**: In Step 3 of Algorithm 1 it says the learned reward predictor predicts rtask- how can it predict that if it was learnt during step 1 and 2 in the absence of any downstream tasks?
> >
> > **R13**: The reward model is trained during Steps 1 and 2, learning to map states and actions to rewards using collected data (states, actions, and rewards). In downstream task learning, the state and action inputs are fed into the reward model to predict rewards. Thus, for different tasks, the model provides the necessary reward information to guide policy learning effectively.
> >
> > > **Q14**: And why are the ri in the transitions collected in line 2 ignored?
> >
> > **R14**: Rewards are not required for dynamics model learning. The objective of the dynamics model is to learn state transitions given different states and actions. Once both the dynamics model and the reward model are trained, they can be utilized together for downstream task learning.
> >
> > > **Q15**: Step 3 of section 4 implies that the causal model is only used to generate curiosity intrinsic rewards (which does not rely on the learned reward predictor at all) so this is inconsistent with Algorithm 1.
> >
> > **R15**: The causal model is not only used to generate curiosity and intrinsic rewards. The causal model serves as the foundation for model-based RL policy learning by predicting state transitions (Eq.(2)) based on state and action inputs. During task learning, it acts as a world model to provide next-state predictions. In ECL, the causal model is further used to generate curiosity-driven intrinsic rewards, which help mitigate overfitting and enhance exploration.
> >
> > **We hope that these comments have addressed the reviewer’s concern. We are happy to answer any follow-up concerns. We thank the reviewer's comments, this helps us a lot to improve the paper!**

---

> > > ### Comment · Reviewer_8bod · 2024-11-25
> > >
> > > Thank you for the clarifications, they have been really helpful for my understanding of your work. I still have some remaining questions/concerns:
> > >
> > > >If the mask is suboptimal in Step 1, the empowerment policy can implicitly refine the causal structure by iteratively maximizing the controllability of the environment while updating the causal mask
> > >
> > > My concern is that it will be difficult to update the causal mask effectively if it is based on an incorrect dynamics encoder, which never gets updated.
> > >
> > > > We have performed the sensitivity analysis on the hyperparameter $\lambda$ in the curiosity reward function across four threshold settings
> > >
> > > Interesting, it looks like adding the curiosity reward (lambda=1) is not significantly better than no curiosity reward (lambda=0) would it not be better to remove it, if it adds complexity with no performance gains (and even harms performance if not scaled carefully)?
> > >
> > > >In downstream task learning, the state and action inputs are fed into the reward model to predict rewards
> > >
> > > But the reward function is often different for downstream tasks, how could it predict them correctly?

---

> ### Author Response · Authors · 2024-11-25
> **Thank you for your responses!**
>
> Thank you so much for your response and for raising these insightful questions. We have provided our discussions below.
>
> > It will be difficult to update the causal mask effectively if it is based on an incorrect dynamics encoder, which never gets updated.
>
> Yes, a good dynamics encoder is generally necessary for learning causal masks. The problem of learning dynamics has been extensively studied in the MBRL literature [1-3]. Recent works in causal MBRL have also integrated structure learning into dynamics models [4-5].
>
> In our work, we jointly learn the dynamics model and causal masks (eq. 5), following a similar approach to [4-5]. Our empirical evaluation demonstrates the ability to learn meaningful causal structures with different causal discovery approaches (see Figures 12-16 in the appendix). Therefore, we believe that learning dynamics is not the primary challenge in this context.
>
> For future work, such as applying ECL to specific domains like real-world robotic systems, task-specific dynamics learning could be incorporated. Potential solutions would be (1) incorporating domain/task/physical knowledge [6]; and (2) pretrain dynamics ensembles with offline data [7]. However, these would be orthogonal to our current work, as the overall framework remains the same—only the method for learning dynamics models would differ.
>
> > It looks like adding the curiosity reward (lambda=1) is not significantly better than no curiosity reward (lambda=0) would it not be better to remove it, if it adds complexity with no performance gains (and even harms performance if not scaled carefully)?
>
> This curiosity reward is used as a regularization term to prevent policy overfitting on causal dynamics, and our results demonstrate performance improvement across tasks. However, you are right that due to the relatively simple causal relationships in these chemical tasks, the performance differences from shaped rewards for preventing overfitting appear modest.
>
> We are actually now conducting additional experiments exploring the impact of this term under complex environments (Manipulation tasks) and also more $\lambda$ values including {0.2; 0.4; 0.6; 0.8}. We will update our results with these findings in this thread once the experiments are complete.
>
> > In downstream task learning, the state and action inputs are fed into the reward model to predict rewards
> But the reward function is often different for downstream tasks, how could it predict them correctly？
>
> Yes, we learn task-specific reward models, similar to approaches in model-based and causal reinforcement learning [1,3,4,5]. This is necessary as each task has its own uniquely defined reward model. Even within the same environment, the reward structure can vary significantly depending on the specific task objectives.
> Hence, for downstream tasks, we follow the approaches in MBRL that involve learning a predictor using state-action pairs to predict rewards from collected data, as described in Section 3.1.
>
> Thank you once again for your response. Please do not hesitate to let us know if you have any further questions!
>
>
> ----
>
> ### References
>
>
> [1] Moerland, Thomas M., et al. "Model-based reinforcement learning: A survey." Foundations and Trends® in Machine Learning 16.1 (2023): 1-118.
>
>
> [2] Deisenroth, Marc, and Carl E. Rasmussen. "PILCO: A model-based and data-efficient approach to policy search." ICML 2011.
>
>
> [3] Hafner, Danijar, et al. "Mastering atari with discrete world models." ICLR 2021.
>
>
> [4] Wang, Zizhao, et al. "Causal dynamics learning for task-independent state abstraction." ICML 2023
>
>
> [5] Huang, Biwei, et al. "Action-sufficient state representation learning for control with structural constraints." ICML 2022.
>
>
> [6] Lutter, Michael, et al. "Differentiable physics models for real-world offline model-based reinforcement learning." ICRA 2021.
>
>
> [7] Rafailov, Rafael, et al. "Moto: Offline pre-training to online fine-tuning for model-based robot learning." CoRL 2023.

---

> ### Author Response · Authors · 2024-11-25
> **Updates on ablation studies of the curiosity reward in complex environment**
>
> As mentioned in the above response, we have conducted additional ablation studies on this curiosity reward in the manipulation reach task. The learning curves for episodic reward and success rate, shown in **Figure 31 of the revision** and results of episodic reward shown below, indicate that the curiosity reward plays a crucial role in the complex environment.
>
> |                      | manipulation (Reach) |
> |----------------------|----------------------|
> | w/ curiosity reward  | 195.6±15.4           |
> | w/o curiosity reward | 94.7±16.5            |
>
> To summarize, the curiosity reward plays a significant role in the framework. However, you are correct that for simpler tasks (e.g., chemical environments), the improvement may not be very significant—though there is still empirical evidence of gains. However, we would keep this since we are targeting to the complex environments, where causal dynamics might overfit. We appreciate you bringing this up and hope that the new empirical validations address your concerns.

---

> > ### Comment · Reviewer_8bod · 2024-11-27
> >
> > Thank you for your response. My understanding of this paper and confidence in its quality has increased, so I will raise my score.

---

> ### Author Response · Authors · 2024-11-27
> **Thank you!**
>
> Thank you for raising the score—we sincerely appreciate your time and effort!
>
> As a side note, besides the validation shown in the above message, we have also conducted experiments under different  $\lambda$ in the manipulation task. The learning curves of success rate are shown in Figure 31 of the updated revision. The results also indicate the importance of the curiosity reward.
>
> Thank you once again for your valuable suggestions. Your suggestions and comments have been invaluable in improving our work.

---

### Official Review · Reviewer_YL1Y · 2024-11-04

**Soundness:** 3
**Presentation:** 3
**Contribution:** 2
**Rating:** 5
**Confidence:** 3

**Summary:**

The paper presents a framework called Empowerment through Causal Learning (ECL), designed to integrate empowerment with causal reasoning in model-based reinforcement learning. ECL operates by training a causal dynamics model, maximizing empowerment under this structure, and updating the model through data gathered from exploration.

**Strengths:**

1. **Detailed Experimental Validation**: The framework is evaluated extensively across multiple environments, including both state-based and pixel-based tasks, showcasing its adaptability and effectiveness in real-world scenarios.
2. **Clear Presentation**: The paper is well-organized and clearly presents concepts, making it accessible and allowing readers to follow the progression of ideas and experimental setups with ease.

**Weaknesses:**

1. **Minor Contribution**: The current framework appears more like a combination of existing approaches rather than a novel advancement. Causal structure learning in model-based RL has been extensively studied in prior work, such as [1-2], as has empowerment in RL [3-4]. This may limit the perceived originality of the contribution, as it builds on established methodologies without significantly advancing them.

2. **High Computational Cost**: The framework’s iterative process of empowerment maximization and causal model updating may result in substantial computational requirements, potentially limiting its scalability in large or dynamic environments.

[1] Huang B, Feng F, Lu C, et al. Adarl: What, where, and how to adapt in transfer reinforcement learning[J]. arXiv preprint arXiv:2107.02729, 2021.

[2] Huang B, Lu C, Leqi L, et al. Action-sufficient state representation learning for control with structural constraints[C]//International Conference on Machine Learning. PMLR, 2022: 9260-9279.

[3] Zhang J, Wang J, Hu H, et al. Metacure: Meta reinforcement learning with empowerment-driven exploration[C]//International Conference on Machine Learning. PMLR, 2021: 12600-12610.

[4] de Abril I M, Kanai R. A unified strategy for implementing curiosity and empowerment driven reinforcement learning[J]. arXiv preprint arXiv:1806.06505, 2018.

**Questions:**

1. As shown in Figure 2, what is the collection policy $\pi_{collect}$, and how do the authors gather the initial dataset (s, a, r, ...) in the buffer?

2. In line 231, the authors state, "the dynamics encoder learned in Step 1 remains fixed, allowing for a focused optimization of both the causal structure and the empowerment in an alternating manner." I am wondering how the authors can fix the optimization of the encoder while still optimizing the causal structure, as shown in Eq. (5)?

3. The proposed framework optimizes iteratively. How is the iteration cycle determined? Will this approach result in high computational costs and longer training times? Could the authors also provide a comparison of training times?

---

> ### Author Response · Authors · 2024-11-20
> **Response to Reviewer YL1Y (1/2)**
>
> We thank the reviewer for the insightful and useful feedback, please see the following for our response.
>
> > **Q1**:  Minor Contribution: The current framework appears more like a combination of existing approaches rather than a novel advancement. Causal structure learning in model-based RL has been extensively studied in prior work, such as [1-2], as has empowerment in RL [3-4]. This may limit the perceived originality of the contribution, as it builds on established methodologies without significantly advancing them.
>
> **R1**: Thank you for raising this concern. Striking a balance between explicit causal discovery and prediction performance, as in [1, 2], is indeed challenging, and the resulting policies often exhibit limited controllability over the system. Similarly, while empowerment encourages efficient exploration, it does not take into account the causal relationships within the environment [3, 4].
>
> Our work introduces the causal empowerment framework, which uniquely integrates considerations of both controllability and causality. Importantly, this is not a simple combination of causal structure learning and empowerment. Instead, our framework addresses critical gaps in existing causal RL approaches, which often fail to fully exploit learned structures for enabling agents to actively explore, enhance system controllability, and utilize collected data to refine causal structures.
>
> Empowerment gain is a natural choice to enhance controllability, and our framework builds upon this by employing an iterative process between empowerment and causal discovery. Furthermore, our approach is method-agnostic, meaning it can seamlessly integrate with existing models, such as those in [1, 2]. This flexibility makes our framework broadly applicable across various scenarios and use cases.
> We sincerely appreciate your perspective and hope this distinction clarifies the novelty and significance of our contribution!
>
> [1] Huang B, Feng F, Lu C, et al. Adarl: What, where, and how to adapt in transfer reinforcement learning[J]. arXiv preprint arXiv:2107.02729, 2021.
>
> [2] Huang B, Lu C, Leqi L, et al. Action-sufficient state representation learning for control with structural constraints[C]//International Conference on Machine Learning. PMLR, 2022: 9260-9279.
>
> [3] Zhang J, Wang J, Hu H, et al. Metacure: Meta reinforcement learning with empowerment-driven exploration[C]//International Conference on Machine Learning. PMLR, 2021: 12600-12610.
>
> [4] de Abril I M, Kanai R. A unified strategy for implementing curiosity and empowerment driven reinforcement learning[J]. arXiv preprint arXiv:1806.06505, 2018.
>
> > **Q2**:  High Computational Cost: The framework’s iterative process of empowerment maximization and causal model updating may result in substantial computational requirements, potentially limiting its scalability in large or dynamic environments.
>
> **R2**: We conducted a detailed analysis of the computation time (h) across two environments (Chain and collider), results are shown below. We can find that ECL achieves its performance improvements with minimal additional computational burden - specifically less than a 10% increase compared to CDL and REG. These findings demonstrate that ECL’s enhanced performance does not come at the expense of significant computational overhead. All experiments were conducted on the same computing platform with the same computational resources. For the detailed analysis, please refer to Appendix D.7 of the revision.
>  | Method | chain | collider |
> |--------|-------|----------|
> | ECL-C  | 32    | 31.5     |
> | CDL    | 30    | 30       |
> | ECL-S  | 30    | 29.6     |
> | REG    | 28    | 28       |
>
> Furthermore, for large-scale environments and downstream tasks, such as real-world robotic applications, we can incorporate pre-trained visual dynamics functions or object-centric representations [5–7] as surrogates. This approach can significantly reduce the computational burden of training the representation and dynamics model. As these are not the focus of our current work, we plan to explore this direction in future work (see the revised future work section).
>
> [5] Shi, Junyao, et al. "Composing Pre-Trained Object-Centric Representations for Robotics From" What" and" Where" Foundation Models." arXiv preprint arXiv:2404.13474 (2024).
>
> [6] Wang, Jianren, et al. "Manipulate by seeing: Creating manipulation controllers from pre-trained representations." Proceedings of the IEEE/CVF International Conference on Computer Vision. 2023.
>
> [7] Luo, Hao, Bohan Zhou, and Zongqing Lu. "Pre-trained Visual Dynamics Representations for Efficient Policy Learning." European Conference on Computer Vision. Springer, Cham, 2024.
>
> **Related Revised Section: Appendix D.7, Section 7 (lines 531-539)**

---

> > ### Author Response · Authors · 2024-11-20
> > **Response to Reviewer YL1Y (2/2)**
> >
> > > **Q3**:  As shown in Figure 2, what is the collection policy $\pi_{collect}$, and how do the authors gather the initial dataset (s, a, r, ...) in the buffer?
> >
> > **R3**:  The policy $\pi_{collect}$ is trained with a reward function $r=\tanh ( {\textstyle \sum_{j=1}^{d_{\mathcal{S}}}}\log \frac{p(s^j_{t+1}|s_t,a_t)}{p(s^j_{t+1}|\mathrm{PA}_{s^j})} )$. This reward function measures the prediction difference between the dense predictor and the current causal predictor, following the approach described in [8].  This reward motivates taking transitions where the dense predictor is better than the causal predictor, which usually suggests the learned causal graph is inaccurate. Hence, we can cover more state-action pairs to expose causal relationships thoroughly. Specifically, by using this policy to interact with the environment, we can effectively initialize our dataset for causal dynamics learning, ensuring comprehensive coverage of state-action pairs to reveal underlying causal relationships. We have detailed the collection policy in Appendix D.2.1 (lines 1105-1109).
> >
> > [8] Wang, Zizhao, et al. "Causal Dynamics Learning for Task-Independent State Abstraction." International Conference on Machine Learning. PMLR, 2022.
> >
> > **Related Revised Section: Appendix D.2.1 (lines 1105-1109)**
> >
> > > **Q4**: In line 231, the authors state, "the dynamics encoder learned in Step 1 remains fixed, allowing for a focused optimization of both the causal structure and the empowerment in an alternating manner." I am wondering how the authors can fix the optimization of the encoder while still optimizing the causal structure, as shown in Eq. (5)?
> >
> > **R4**: To learn the causal mask, we first train a robust dynamics encoder by optimizing Eq. (4), maximizing the likelihood of the observed trajectories. Once the encoder is optimized, we fix its parameter $\phi_c$, and proceed to update the causal mask by maximizing $\mathcal{L}\_{\rm{c-dyn}}$  defined in Eq. (5). Hence, the causal structure is refined through the optimization of the causal mask $M$ by maximizing $\mathcal{L}_{\rm{c-dyn}}$  in both dynamics predictor (the first component in Eq. (5)) and causal structure regularization (the second component in Eq. (5)). We have provided clarification in Section 3.2 (line 230-231).
> >
> > **Related Revised Section: Section 3.2 (lines 230-231)**
> >
> > > **Q5**: The proposed framework optimizes iteratively. How is the iteration cycle determined? Will this approach result in high computational costs and longer training times? Could the authors also provide a comparison of training times?
> >
> > **R5**: We determine iteration cycles empirically and we did not specifically tune this parameter. **Importantly, We set fixed epoch numbers for each stage across environments.** The detailed training step is detailed in Tables 2 and 4 of Appendix D.2 (line 1126 and line 1162).
> >
> > |                   | Chemical | Physical | Manipulation |
> > |-------------------|----------|----------|--------------|
> > | dynamics learning | 500K     | 500K     | 32M          |
> > | task learning     | 300K     | 1.5M     | 2M           |
> >
> > Regarding computational costs, we have calculated and compared the training times across different methods. As mentioned in the response to Q2, ECL achieves its performance improvements with minimal additional computational costs—less than 10% overhead compared to CDL or REG. For a detailed breakdown, please refer to Appendix D.7 in the revised paper.
> >
> > **Related Revised Section: Appendix D.2 (line 1126 and line 1162) and D.7**
> >
> > We hope that these comments have addressed the reviewer’s concern. We are happy to answer any follow-up concerns. We thank the reviewer's comments, this helps us a lot to improve the paper!

---

> > > ### Author Response · Authors · 2024-11-25
> > >
> > > Thank you once again for your time and effort in reviewing our paper and for providing such helpful comments! We have addressed the points you raised, with particular attention to clarifying our contributions (**R1**), conducting additional evaluations including computational analysis (**R2**) and hyperparameter ablation studies (**R5**), and providing further clarifications on our method (**R3** and **R4**). All corresponding clarifications have been updated into the revised version of the main paper and appendix (with changes marked in red and cross-referenced in the response above).
> > >
> > > As the deadline approaches, please do not hesitate to let us know if you have any additional comments or concerns. Thank you again for your valuable feedback and effort!

---

> > > > ### Author Response · Authors · 2024-11-27
> > > > **Thank you and we are happy to take further questions if any**
> > > >
> > > > Dear Reviewer YL1Y,
> > > >
> > > > Thank you once again for your insightful review. We have provided a detailed response along with related discussions and revisions in the paper.
> > > >
> > > > With the revision deadline approaching in a few hours, please feel free to let us know if you have any further questions or concerns. We will be happy to address them in the revision before the deadline. Alternatively, we would also like to discuss them in this thread before the discussion phase closes.
> > > >
> > > > We sincerely appreciate your time, effort, and valuable comments.
> > > >
> > > > Thanks and regards,
> > > >
> > > > Submission 1429 Authors

---

> > > > > ### Comment · Reviewer_YL1Y · 2024-12-02
> > > > > **Thank you for your response**
> > > > >
> > > > > Thank you to the authors for their response. I appreciate the efforts to provide additional experimental results. However, I still have concerns that since the iteration cycles are determined empirically, it may be challenging to extend this method to real-world scenarios.

---

> ### Author Response · Authors · 2024-12-02
> **Thank you for your feedback! We have provided further clarification here**
>
> Thank you for your feedback!
>
> To clarify, the iteration cycles in our framework do not require fine-tuning across different environments or tasks. Instead, we use fixed  numbers of the iteration across diverse experiments for each domain, including different experiments in chemical, physical, and manipulation tasks. This demonstrates that our framework is not overly sensitive to this hyperparameter.
>
> And thanks for raising the point of real-world scenarios. For those real-world deployment, the number of iteration steps can be determined based on task-specific factors such as task complexity, # steps used for standard policy learning. Additionally, monitoring metrics like loss magnitudes or rewards can be also useful for setting how many steps of iterations we need to use, similar to the generic and typical deployment process for RL in real-world cases like robot learning. We would like to point out that addressing real-world, large-scale scenarios remains the focus of our future work. And, in the current paper, we have tested our approach on well-established RL benchmarks without significantly tuning this parameter, further confirming that it does not pose a specific challenge (compared with those baselines).
>
> Thank you again for your response! If you have any further questions, please feel free to let us know. We are more than happy to provide further discussions.

---

### Author Response · Authors · 2024-11-22
**General Response**

We sincerely appreciate the thoughtful feedback and insightful comments from all reviewers. Your constructive suggestions have been invaluable in improving our work's clarity and identifying key areas for deeper analysis. We have addressed each point in individual rebuttals and revised our paper **(marked in red)** accordingly.

We are grateful that the reviewers consider our paper's contribution "novel and interesting" (Reviewers ```8bod, 8gmf, DR5q```), presentation "well-organized, clearly presents concepts" (Reviewer ```YL1Y```) and "clear and easy to follow" (Reviewers ```DR5q```), empirical evaluations include "adaptability and effectiveness" (Reviewer ```YL1Y```), "wide spread of environments/diverse set of environments" (Reviewer ```8bod```), “enough details and explanations” (Reviewers ```8gmf, DR5q```), and ”strong performance” (Reviewer ```8gmf```).

Here we summarize the common concerns along with our responses and the corresponding modifications, providing additional context to further clarify our approach.

### Clarification of Technical Details

* Novelty of the proposed framework in [```R1```, Reviewer ```YL1Y```].

* Data collection policy in [```R3```, Reviewer ```YL1Y```], causal model learning in [```R4```, Reviewer ```YL1Y```; ```R4-R9, R11, R13-15```, Reviewer ```8bod``` ].

* Task learning processes in [```R1```, Reviewer ```8bod```] and  [```Section 3.1```, ```3.2```, ```4``` and ```Appendix D.2```].

* Detailed clarification of notations of the essential variables and parameters in the objective functions in  [```R1```, Reviewer ```4AH7```] and [```Section 2```, ```Table 2```, ```Appendix C.1```].

* Illustration of the trainable parameters in each objective function in [```R4```, Reviewer ```YL1Y```; ```R3```, Reviewer ```4AH7```] and [```Appendix C.2```].

### More Evaluation

* Computational cost analysis in [```R2```, Reviewer ```YL1Y```] and [```Appendix D.7```].

* Ablation studies on causality, curiosity, and empowerment-driven exploration in [```R10```, Reviewer ```8bod```; ```R2```, Reviewer ```8gmf```; ```R3```, Reviewer ```DR5q```]  and [```Appendix D.8```].

* More comparison against SOTA MBRL algorithms [```R3```, Reviewer ```8bod```] and [```Appendix D.6```].

* Clarification of the experimental setup in [```R5```, Reviewer ```YL1Y```; ```R2```, Reviewer ```8bod```; ```R5```, Reviewer ```4AH7```]  and results in [```R12```, Reviewer ```8bod```; ```R1```, Reviewer ```8gmf```; ```R4```, Reviewer ```DR5q```] and [```Section 5.2.1```, ```Appendix D.2.1``` and ```Appendix D.6```].

We hope that our detailed responses have addressed the reviewer’s concern. We are happy to answer any follow-up concerns. We thank you again for your valuable time, effort, and expertise!

---

### Meta-Review · Area_Chair_yqhF · 2024-12-19

**Metareview:**

The reviewers all agree this paper provides a clear presentation of a novel approach to incorporating causal structure into empowerment-driven exploration. I encourage the authors to further refine their language and discussion to maximize clarity for the camera-ready version of this paper.

**Additional Comments On Reviewer Discussion:**

Reviewers mainly asked for clarifying details around the method and experiment design, on which the authors provided additional details in their rebuttals.

---

### Decision · Program_Chairs · 2025-01-22

Accept (Poster)